# Analysis of the immune response to sciatic nerve injury identifies efferocytosis as a key mechanism of nerve debridement

Ashley L Kalinski[1†], Choya Yoon[1†], Lucas D Huffman[1,2†], Patrick C Duncker[3†], Rafi Kohen[1,2], Ryan Passino[1], Hannah Hafner[1], Craig Johnson[1], Riki Kawaguchi[4], Kevin S Carbajal[3], Juan Sebastian Jara[5], Edmund Hollis[5,6], Daniel H Geschwind[4], Benjamin M Segal[7,8], Roman J Giger[1,2,3]*

[1]Department of Cell and Developmental Biology, University of Michigan Medical School, Ann Arbor, United States; [2]Neuroscience Graduate Program, University of Michigan Medical School, Ann Arbor, United States; [3]Department of Neurology, University of Michigan Medical School, Ann Arbor, United States; [4]Program in Neurogenetics, Department of Neurology, David Geffen School of Medicine, University of California, Los Angeles, Los Angeles, United States; [5]Burke Neurological Institute, White Plains, United States; [6]The Feil Family Brain and Mind Research Institute, Weill Cornell Medicine, New York, United States; [7]Department of Neurology, The Ohio State University Wexner Medical Center, Columbus, United States; [8]The Neurological Institute, The Ohio State University, Columbus, United States

*For correspondence:
rgiger@umich.edu

[†]These authors contributed equally to this work

Competing interests: The authors declare that no competing interests exist.

**Abstract** Sciatic nerve crush injury triggers sterile inflammation within the distal nerve and axotomized dorsal root ganglia (DRGs). Granulocytes and pro-inflammatory Ly6C$^{high}$ monocytes infiltrate the nerve first and rapidly give way to Ly6C$^{negative}$ inflammation-resolving macrophages. In axotomized DRGs, few hematogenous leukocytes are detected and resident macrophages acquire a ramified morphology. Single-cell RNA-sequencing of injured sciatic nerve identifies five macrophage subpopulations, repair Schwann cells, and mesenchymal precursor cells. Macrophages at the nerve crush site are molecularly distinct from macrophages associated with Wallerian degeneration. In the injured nerve, macrophages 'eat' apoptotic leukocytes, a process called efferocytosis, and thereby promote an anti-inflammatory milieu. Myeloid cells in the injured nerve, but not axotomized DRGs, strongly express receptors for the cytokine GM-CSF. In GM-CSF-deficient ($Csf2^{-/-}$) mice, inflammation resolution is delayed and conditioning-lesion-induced regeneration of DRG neuron central axons is abolished. Thus, carefully orchestrated inflammation resolution in the nerve is required for conditioning-lesion-induced neurorepair.

## Introduction

In the injured adult mammalian CNS, the regenerative capacity of severed axons is very limited. However, regeneration of dorsal column axons in the rodent spinal cord can be augmented if preceded by a conditioning lesion to the sciatic nerve (*McQuarrie et al., 1977*; *Neumann and Woolf, 1999*; *Richardson and Issa, 1984*). This seminal observation has been exploited extensively to identify mechanisms that promote axon regeneration (*Abe and Cavalli, 2008*; *Blesch et al., 2012*; *Chandran et al., 2016*). Traumatic PNS injury leads to sterile inflammation at the site of injury and within the distal nerve stump where axons undergo Wallerian degeneration (*Kim and Moalem-Taylor, 2011*; *Perry et al., 1987*). In addition, a remote inflammatory response is observed in

axotomized dorsal root ganglia (DRGs) (*Hu and McLachlan, 2003*; *Lu and Richardson, 1991*) and the lumbar spinal cord (*Guan et al., 2016*; *Hu et al., 2007*; *Zhang et al., 2007*). The innate arm of the immune system is important for peripheral nerve regeneration, as well as conditioning-lesion-induced dorsal column axon regeneration (*Kwon et al., 2015*; *Niemi et al., 2013*; *Salegio et al., 2011*; *Zigmond and Echevarria, 2019*). Very recent studies employed single-cell RNA sequencing (scRNA-seq) to describe gene expression in naive and injured peripheral nervous tissue at cellular resolution (*Wang et al., 2020*; *Wolbert et al., 2020*; *Ydens et al., 2020*). A comparative analysis of immune cell profiles within the injured sciatic nerve and axotomized DRGs, however, has not yet been carried out.

The sciatic nerve trunk is covered by the epineurium, a protective connective tissue sheath that harbors fibroblasts, macrophages, and blood vessels. The more delicate perineurium covers nerve bundles and the endoneurium is a tube-like structure wrapped around individual myelinated fibers. The endoneurium contains macrophages and fibroblast-like mesenchymal cells (MES) (*Carr et al., 2019*; *Ydens et al., 2020*). Following PNS injury, Schwann cells (SC) reprogram into repair cells and together with MES and nerve-resident macrophages produce chemokines and cytokines to promote entry of hematogenous immune cells (*Arthur-Farraj et al., 2012*; *Müller et al., 2010*; *Richard et al., 2012*; *Ydens et al., 2020*). Repair SC, together with innate immune cells, contribute to nerve debridement, formation of new blood vessels, and release of growth promoting molecules, thereby creating a microenvironment conducive for long-distance axon regeneration and tissue repair (*Barrette et al., 2008*; *Clements et al., 2017*; *DeFrancesco-Lisowitz et al., 2015*; *Höke et al., 2000*; *Martini et al., 2008*). Despite recent progress, it remains unclear which cell types in the injured nerve contribute to tissue debridement and there is a paucity in our understanding of the underlying molecular mechanisms (*Brosius Lutz et al., 2017*; *Klein and Martini, 2016*).

Sciatic nerve injury leads to a remote and strong cell body response in axotomized DRG neurons (*Chandran et al., 2016*). This includes induction of neuron-intrinsic growth programs, neuronal release of cytokines and chemokines, activation of intra-ganglionic tissue resident macrophages, immune-like glia, and entry of hematogenous leukocytes (*Cafferty et al., 2004*; *McLachlan and Hu, 2014*; *Richardson and Lu, 1994*; *Richardson et al., 2009*; *Wang et al., 2018*; *Zigmond and Echevarria, 2019*). Experimentally induced intra-ganglionic inflammation, triggered by injection of *C. parvum* bacteria into DRGs, increases axon regeneration following dorsal root injury (*Lu and Richardson, 1991*). Intra-ganglionic expression of recombinant CCL2 leads to increased macrophage staining, enhanced DRG neuron outgrowth in vitro (*Niemi et al., 2016*), and regeneration of DRG neuron central projections following spinal cord injury (*Kwon et al., 2015*).

Here, we employed a combination of flow cytometry, mouse reporter lines, and immunofluorescence labeling to describe the leukocyte composition in the injured sciatic nerve and axotomized DRGs. We used parabiosis to show that upon sciatic nerve crush injury (SNC), the origin, magnitude, and cellular composition of immune cell profiles is very different between the nerve and DRGs. For a comparative analysis, we carried out bulk RNA sequencing of DRGs and single-cell RNA sequencing (scRNA-seq) of injured nerves. We report the cellular make up, cell-type-specific gene expression profiles, and lineage trajectories in the regenerating mouse PNS. Computational analysis revealed cell-type-specific expression of engulfment receptors and bridging molecules important for eating of apoptotic cell corpses, a process called efferocytosis (*Henson, 2017*). We show that within the injured nerve, monocytes (Mo) and macrophages (Mac) eat apoptotic leukocytes, and thus, contribute to inflammation resolution. Strikingly, Mac at the nerve injury site are molecularly distinct from Mac in the distal nerve stump. *Csf2ra* and *Csf2rb*, obligatory components of the GM-CSF receptor (*Hansen et al., 2008*), are strongly expressed by myeloid cells in the injured nerve, but not in axotomized DRGs. Functional studies with *Csf2*$^{-/-}$ mice, deficient for GM-CSF, show that this cytokine regulates the inflammatory milieu in the injured nerve and is important for conditioning-lesion-elicited dorsal column axon regeneration. Taken together, our work provides novel insights into a rich and dynamic landscape of injury-associated cell states, and underscores the importance of properly orchestrated inflammation resolution in the nerve for neural repair.

## Results

### Quantitative analysis of immune cell profiles in the injured sciatic nerve

Despite recent advances in our understanding of PNS injury-induced inflammation, a comparative analysis of the leukocyte subtypes within the injured sciatic nerve and axotomized DRGs does not yet exist. For identification and quantification of immune cell profiles at different post-injury time points, adult mice were subjected to a mid-thigh sciatic nerve crush (SNC) injury. SNC leads to axon transection, but preserves the surrounding epineurium (*Figure 1A*). Flow cytometry was used to assess the composition of injury-mobilized immune cell profiles in the nerve and DRGs (gating strategy is illustrated in *Figure 1—figure supplement 1*). To minimize sample contamination with circulating leukocytes, mice were perfused with physiological saline prior to tissue collection. The nerve trunk was harvested and divided into a proximal and distal segment. The distal segment included the injury site together with the distal nerve stump (*Figure 1A*). For comparison, the corresponding tissues from naive mice were collected. In naive mice, ~300 live leukocytes (CD45$^+$) are detected within a ~ 5 mm nerve segment. At day 1 following SNC (d1), the number of CD45$^+$ cells in the distal nerve increases sharply, peaks around 23,100 ± 180 cells at d3, and declines to 14,000 ± 200 at d7 (*Figure 1B*). Further analysis shows that granulocytes (GC), identified as CD45$^+$CD11b$^+$Ly6G$^+$CD11c$^-$ cells, are absent from naive nerve, but increase to 7,800 ± 300 at d1. By d3, the number of GC dropped below 1000 (*Figure 1C*). A robust and prolonged increase of the Mo/Mac population (CD45$^+$CD11b$^+$Ly6G$^-$CD11c$^-$) is observed, reaching 7300 ± 120 cells at d1, peaking around 13,200 ± 240 at d3, and declining to 3200 ± 90 at d7 (*Figure 1D*). Monocyte-derived dendritic cells (MoDC), identified as CD45$^+$CD11b$^+$Ly6G$^-$CD11c$^+$ cells, increase more gradually. They are sparse at d1, reach 1100 ± 30 at d3, and 3400 ± 60 at d7 (*Figure 1E*). Few CD11b$^-$ conventional DC (cDC), identified as CD45$^+$CD11b$^-$Ly6G$^-$CD11c$^+$ cells, are present at d1 and d3 and cDC increase to 600 ± 20 at d7 (*Figure 1F*). The total number of lymphocytes (CD45$^+$CD11b$^-$CD11c$^-$Ly6G$^-$) is low, but significantly elevated at d1, d3, and d7 post-SNC (*Figure 1G and H*). In marked contrast to the distal nerve stump, flow cytometry of the proximal nerve stump shows that SNC does not significantly alter immune cells number or composition (*Figure 1—figure supplement 2A–K*). The sharp divide in myeloid cell distribution within the injured nerve is readily seen in longitudinal sections stained with anti-F4/80 (*Figure 1—figure supplement 2L*). The distal nerve stump was identified by anti-GFAP staining, a protein upregulated in repair Schwann cells (*Figure 1—figure supplement 2L*). In sum, SNC-elicited inflammation in the nerve is confined to the crush site and the distal nerve stump where severed fibers undergo rapid Wallerian degeneration. GC increase sharply and peak within 24 hr, followed by Mo/Mac, MoDC, and few lymphocytes.

### Quantitative analysis of immune cell profiles in axotomized DRGs

Immunofluorescence staining of DRG sections shows that SNC causes a transient increase in Iba1 and F4/80 immunolabeling, peaking around d3 and declining at d7 (*Figure 2A*). Flow cytometric analysis of DRGs from naive mice identifies on average ~600 live leukocytes per ganglion, including GC, Mo/Mac, MoDC, cDC, and lymphocytes (*Figure 2B–G*). At d1, no significant change in intraganglionic immune cell profiles is observed. At d3, there is a ~ 2-fold increase in leukocytes; however, a significant increase is only observed for Mo/Mac (*Figure 2C*). At d7, the Mo/Mac population is significantly reduced compared to d3. The MoDC and cDC populations are elevated at d7 when compared to DRGs from naïve mice (*Figure 2D and E*). Lymphocytes are present in naive DRGs but do not significantly increase during the first week post-SNC (*Figure 2F and G*). The presence of CD3$^+$ T cells in DRGs was validated by immunofluorescence labeling of L5 DRG sections (*Figure 2—figure supplement 1*). For an independent assessment of the kinetics and magnitude of SNC-induced inflammation in the nerve trunk and DRGs, we used western blotting to carry out a 3-week time-course analysis. Probing tissue lysates with anti-CD11b shows that the injury induced increase in myeloid cells in the nerve trunk exceeds the one in axotomized DRGs by an order of magnitude (*Figure 2H and I*). Taken together, these studies show that SNC induces a remote immune response in axotomized DRGs that is strikingly different in magnitude and cellular composition from injured nerve tissue.

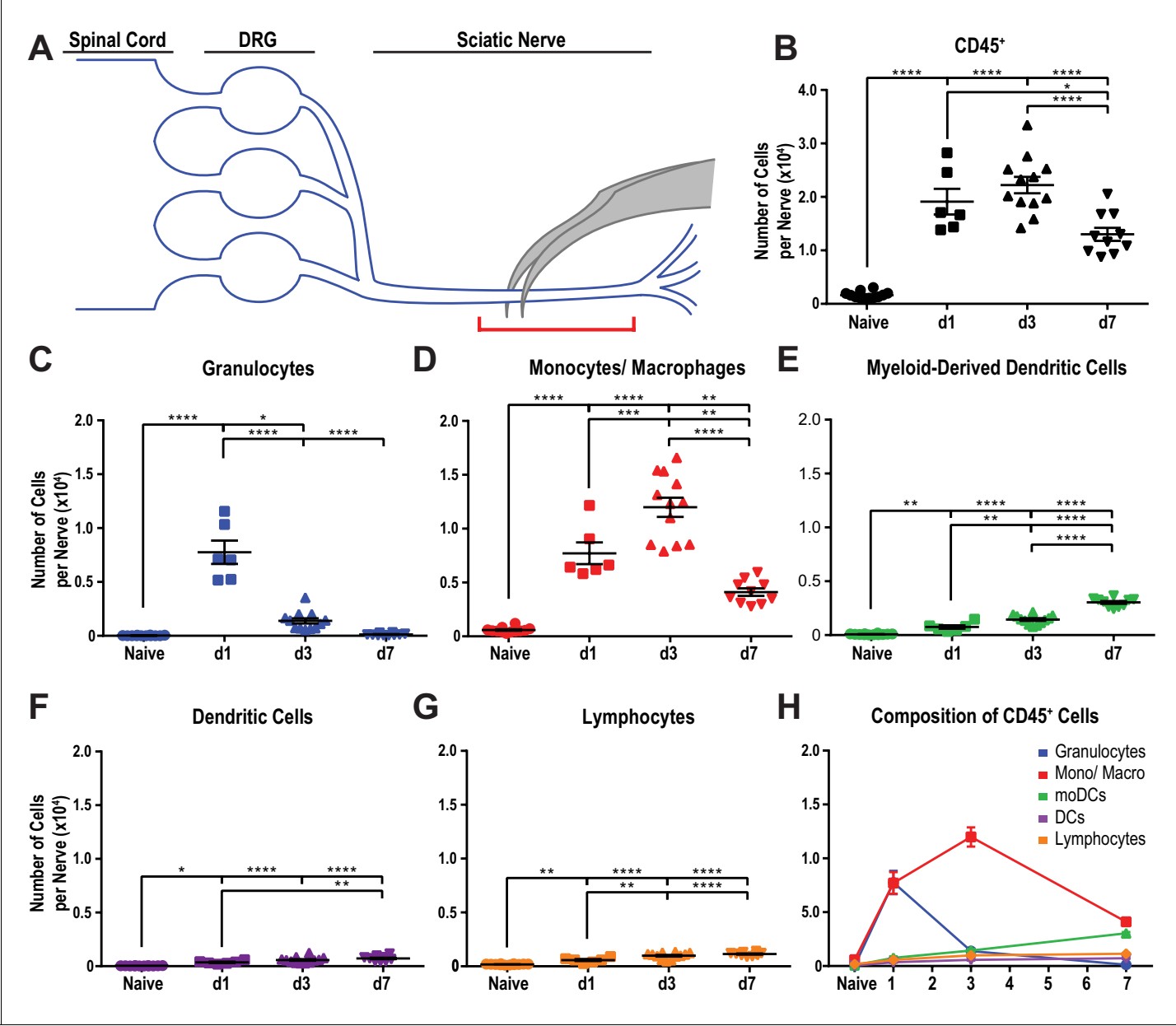

**Figure 1.** Immune cell profiles in the injured murine sciatic nerve. (A) Anatomy of lumbar spinal cord and DRGs connected to the sciatic nerve. The location of the crush site within the nerve trunk and the tissue segment collected for flow cytometry (red bracket) are shown. (B) Quantification of live, CD45$^+$ leukocytes, normalized per sciatic nerve trunk. Flow cytometry of nerve tissue collected from naive mice (n = 10 biological replicates, with six nerves per replicate), day 1 (d1) post-SNC (n = 7), d3 (n = 12), and d7 (n = 12). (C). Quantification of granulocytes (CD45$^+$, CD11b$^+$, Ly6G$^+$) per nerve trunk. (D). Quantification of Mo/Mac (CD45$^+$, CD11b$^+$, CD11c$^-$, Ly6G$^-$) per nerve trunk. (E). Quantification of MoDC (CD45$^+$, CD11b$^+$, CD11c$^+$, Ly6G$^-$) per nerve trunk. (F) Quantification of cDC (CD45$^+$, CD11b$^-$, CD11c$^+$, Ly6G$^-$) per nerve trunk. (G). Quantification of lymphocytes (CD45$^+$, CD11b$^-$) per nerve trunk. (H). Composition of CD45$^+$ leukocytes in the nerve trunk at different post-injury time points. Flow data are represented as mean cell number ± SEM. Statistical analysis was performed in GraphPad Prism (v7) using one-way or two-way ANOVA with correction for multiple comparisons with Tukey's post-hoc test. For B-G, unpaired two-tailed t-test with Welch's correction. A p value < 0.05 (*) was considered significant. p<0.01 (**), p<0.001 (***), and p<0.0001 (****).

The online version of this article includes the following figure supplement(s) for figure 1:

**Figure supplement 1.** Gating scheme for flow cytometry.
**Figure supplement 2.** Immune cell profiles in the sciatic nerve proximal to the injury site.

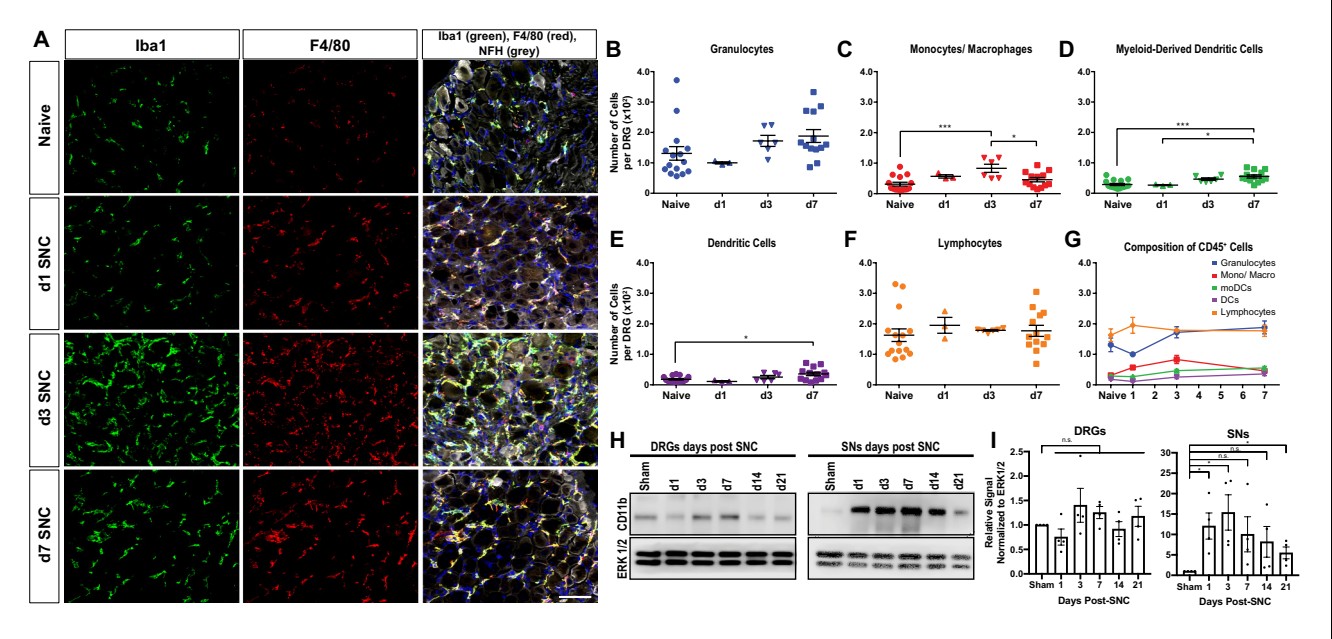

**Figure 2.** Immune cell profiles in axotomized DRGs. (**A**) Representative images of L4 DRG cross sections from naïve mice, d1, d3, and d7 post-SNC. Macrophages were stained with anti-Iba1 and anti-F4/80. Neurons were stained with anti-NFH. Scale bar, 50 µm. (**B**) Quantification of granulocytes per DRG detected by flow cytometry. For flow cytometry of DRGs, naïve mice (n = 14 biological replicates), d1 (n = 3), d3 (n = 5), and d7 (n = 12) post-SNC mice were used. Granulocytes (CD45$^+$, CD11b$^+$, Ly6G$^+$) per DRG are shown. (**C**) Quantification of Mo/Mac (CD45$^+$, CD11b$^+$, CD11c$^-$, Ly6G$^-$) per DRG. (**D**) Quantification of MoDC (CD45$^+$, CD11b$^+$, CD11c$^+$, Ly6G$^-$) per DRG. (**E**) Quantification of cDC (CD45$^+$, CD11b$^-$, CD11c$^+$, Ly6G$^-$) per DRG. (**F**) Quantification of lymphocytes (CD45+, CD11b$^-$) per DRG. (**G**) Composition of CD45$^+$ leukocytes in lumbar DRGs identified by flow cytometry. Flow data are represented as mean cell number ± SEM. Each data point represents L3-L5 DRGs pooled from three to four animals (18–24 DRGs), biological replicates, n = 3–14. Statistical analysis was performed in GraphPad Prism (v7) using one-way or two-way ANOVA with correction for multiple comparisons with Tukey's post-hoc test. For B-F, unpaired two-tailed t-test with Welch's correction. A p value < 0.05 (*) was considered significant. p<0.01 (**), p<0.001 (***), and p<0.0001 (****). (**H**) Western blots analysis of DRGs and sciatic nerves (SNs) prepared from sham operated mice and mice at different post-SNC time points (d1–d21), probed with anti-CD11b and anti-ERK1/2 as loading control. (**I**) Quantification of CD11b signal in DRGs and SNs. Unpaired two-tailed Student's t-test compared to sham operated. n.s. not significant, *p<0.05, biological replicates n = 4 (with four mice for each time point).

The online version of this article includes the following figure supplement(s) for figure 2:

**Figure supplement 1.** T cells in naïve and axotomized DRGs.

## Sciatic nerve injury triggers massive infiltration of immune cells into the injured nerve, but not axotomized DRGs

Endoneurial Mac in the sciatic nerve and DRGs respond to injury (*Mäurer et al., 2003*; *Müller et al., 2010*); however, there are no reliable cell surface markers to distinguish between tissue resident and injury mobilized hematogenous immune cells that enter the nerve or axotomized DRGs. To examine cell origin, we employed parabiosis, that is conjoined wildtype (WT) and tdTomato (tdTom) reporter mice that share blood circulation. We chose parabiosis over bone marrow transplantation because of potential confounding effects caused by irradiation (*Guimarães et al., 2019*). One month after parabiosis surgery, both parabionts were subjected to unilateral SNC. Sciatic nerves, DRGs, and spinal cords were harvested at different post-injury time points (*Figure 3A*). Shared blood circulation was assessed by flow cytometry of the spleen, and revealed a myeloid cell (CD45$^+$CD11b$^+$) chimerism of 27.3 ± 1.5 (gating strategy is illustrated in *Figure 3—figure supplement 1*). At d3 following SNC, flow cytometric analysis of nerves isolated from WT parabionts identifies 28.4 ± 6.7% tdTom$^+$ myeloid (CD45$^+$CD11b$^+$) cells (*Figure 3B*). Fractionation of myeloid cells into Mo/Mac (CD45$^+$CD11b$^+$Ly6G$^-$CD11c$^-$) and MoDC (CD45$^+$CD11b$^+$Ly6G$^-$CD11c$^+$) further revealed that 27.1 ± 6.9% of Mo/Mac and 30 ± 5.6% of MoDC are tdTom$^+$ in the injured WT parabiont (*Figure 3C*). When coupled with ~27% chimerism (*Figure 3—figure supplement 1C*), this suggests that blood-borne cells make up the vast majority of immune cells in the injured nerve. Histological analysis of injured nerves

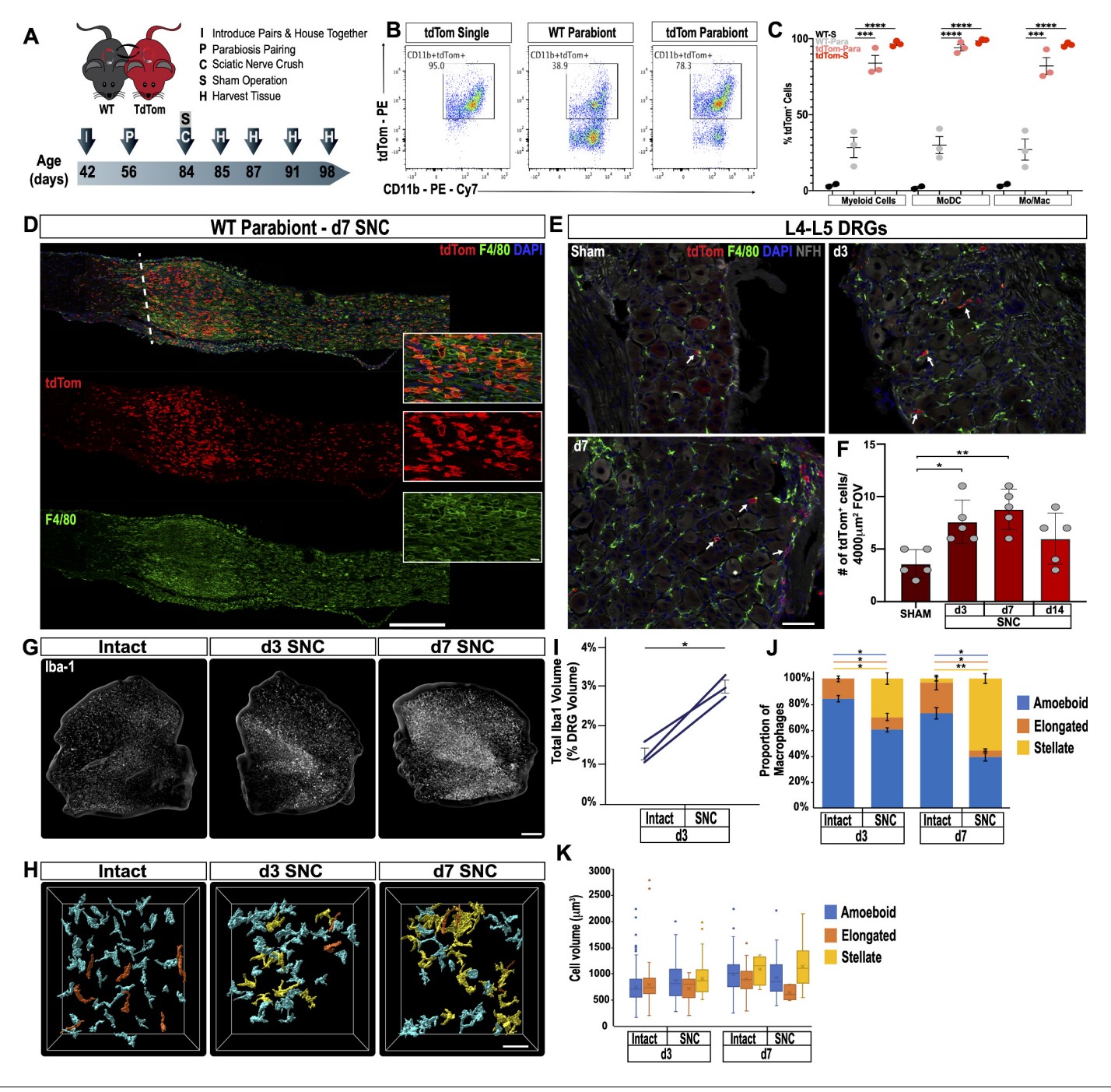

**Figure 3.** Sciatic nerve injury triggers massive accumulation of hematogenous leukocytes in the injured nerve but not axotomized DRGs. (A) Parbiosis complex of a wildtype (WT) and a tdTomato (tdTom) mouse. Mice were surgically paired at postnatal day 56. The timeline of the experiment is shown. (B) Flow cytometric analysis of sciatic nerve trunks collected from non-parabiotic (single) tdTom mice, WT parabionts, and tdTom parabionts. Dotplot of live (CD11b⁺, tdTom⁺) cells in the d3 post-SNC nerve. (C) Quantification of tdTom⁺ myeloid cells in the 3d injured nerve of WT single mice (WT-S), WT parabiont (WT-para), tdTom parabiont (tdTom-para), and tdTom single (tdTom-S) mice. The fraction of tdTom⁺ myeloid cells (CD45⁺, CD11b⁺), MoDC (CD45⁺, CD11b⁺, CD11c⁺, Ly6G⁻), and Mo/Mac (CD45⁺, CD11b⁺, CD11c⁻, Ly6G⁻) is shown. For quantification of tdTom⁺ immune cells, nerves from the WT parabiont and the tdTom parabiont were harvested separately (three mice per data point) with n = 2–3 biological replicates. Flow data are represented as fraction of tdTom⁺ cells ± SEM. Statistical analysis was performed in GraphPad Prism (v8) using one-way ANOVA with correction for multiple comparisons with Tukey's post-hoc test. p value of < 0.001 (***) and p<0.0001 (****). (D) Longitudinal sciatic nerves sections of the WT parabiont at d7 post-SNC. The nerve crush site is marked with a white dotted line, proximal is to the left, distal to the right. Anti-F4/80 (green) and tdTom⁺ cells (red) staining is shown. Scale bar, 500 μm, for insets, 20 μm. (E) Lumbar DRG cross sections of WT parabionts harvested from sham

*Figure 3 continued on next page*

Figure 3 continued

operated mice, at d3, and d7 post-SNC. Sections were stained with anti-F4/80 (green) and anti-NF200 (white). Hematogenous (tdTom$^+$) leukocytes are marked with white arrows. Scale bar, 50 μm. (F) Quantification of tdTom$^+$ cells per field of view (FOV = 4000 μm$^2$) in DRG sections of the WT parabiont. Data are shown as number of tdTom$^+$ cells ± SEM, n = 3–5 mice per time point. Student's t test with p<0.5 (*) considered statistically significant, p<0.01 (**). (G) Whole mount anti-Iba1 immunofluorescence staining of L4 DRGs from intact, d3, and d7 post-SNC time points. Scale bar, 200 μm. (H) Morphological reconstruction of Iba1$^+$ cells in DRGs with Imaris. Analysis of DRG resident macrophages revealed amoeboid (cyan) and elongated (orange) morphologies if the nerve was not injured. At d3 and d7 post-SNC, a subpopulation of Iba1$^+$ cells with stellate (yellow) morphology was observed in DRGs. Scale bar, 50 μm. (I) Quantification of total volume of Iba1$^+$ structures in DRGs, rendered by Imaris. The total volume of Iba1$^+$ structures per DRG was quantified on the intact side and the injured side of the same mouse at d3 post-SNC (n = 3 mice). Paired Students t test, p value < 0.05 (*), was considered significant. (J) Quantification of Iba1$^+$ cells with amoeboid, elongated, and stellate morphologies. (K) Quantification of cell volume of individual Iba1$^+$ cells with amoeboid, elongated, and stellate morphologies. At d3 post-SNC, a total of 416 cells were reconstructed on the intact side and a total of 234 cells on the injured side. At d7 post-SNC, a total of 136 cells were reconstructed on the intact side and a total of 93 cells on the injured side. The distribution of morphological categories ± SEM (J) and cell volumes ± SEM (K) are shown. Paired, two-tailed Student's t test, a p value < 0.05 (*) was considered significant. p<0.01 (**).

The online version of this article includes the following figure supplement(s) for figure 3:

**Figure supplement 1.** Flow cytometry gating scheme to assess chimerism of parabiotic mice.

from WT parabionts identified numerous tdTom$^+$ cells (*Figure 3D*). During the first 24 hr, tdTom$^+$ cells are confined to the injury site (data not shown). At d3 and d7, tdTom$^+$ cells are preferentially found at the injury site but also present within the distal nerve stump where fibers undergo Wallerian degeneration (*Figure 3D* and *Figure 6—figure supplement 2C*). In the proximal nerve, very few tdTom$^+$ cells are detected at any post-SNC time point (*Figure 3D*). A 2-week time course analysis of axotomized DRGs harvested from WT parabionts identified a modest and transient increase of tdTom$^+$ cells (*Figure 3E*). DRG sections from naive mice revealed that the number of tdTom$^+$ cells per field-of-view (4000 μm$^2$) is very low. Following SNC, there is a modest, but statistically significant increase in tdTom$^+$ cells at d3 and d7, but not at 14 days, suggesting that only a small number of hematogenous leukocytes enter axotomized DRGs (*Figure 3F*). Together these studies show that SNC-elicited intra-ganglionic increase of Iba1$^+$ and F4/80$^+$ immune profiles (*Figure 2A*) primarily occurs through mechanisms that involve DRG-resident macrophages, rather than hematogenous immune cells. Of note, during the first 2 weeks post-SNC, no tdTom$^+$ cells were detected in the lumbar spinal cord (data not shown), suggesting that hematogenous immune cells do not significantly contribute to SNC-triggered spinal cord inflammation.

## Sciatic nerve injury triggers macrophage morphological changes in axotomized DRGs

In tissue sections of axotomized DRGs, there is a rapid increase in Iba1 and F4/80 immunoreactive profiles (*Figure 2A*), yet in DRGs of parabiotic mice the number of blood-derived tdTom$^+$ immune cells is modest (*Figure 3E and F*). This raises questions regarding the underlying cellular basis of increased Iba1 immunoreactivity. Previous studies reported that upon sciatic nerve injury, DRG resident Mac undergo limited proliferation (*Leonhard et al., 2002*; *Yu et al., 2020*). To examine whether altered macrophage morphology may contribute to increased Iba1 staining, axotomized DRGs were subjected to whole-mount immunofluorescence labeling with anti-Iba1 (*Figure 3G*). Three-dimensional projection analysis of Mac profiles, in the absence of nerve injury (intact) and at 3 days post-SNC, revealed a 2.3-fold increase in the total volume occupied by Iba1$^+$ cells (*Figure 3I*). Two distinct Mac morphologies were observed in intact DRGs, a majority (84 ± 2%) of amoeboid cells and a smaller population (16 ± 2%) of elongated cells (*Figure 3H and J*). SNC triggers Mac morphological changes in axotomized DRGs (*Figure 3H*). Many Iba1$^+$ cells acquire a more complex, stellate morphology and exhibit enveloping extensions. At d3, Mac with amoeboid (60 ± 2%), elongated (10 ± 3%), and stellate morphologies (30 ± 4%) are identified. And at d7, amoeboid (40 ± 3%), elongated (5 ± 1%), and stellate (55 ± 4%) shaped Mac are detected (*Figure 3J*). While the SNC-triggered Mac morphological changes are quite striking, they do not alter the average volume of individual cells (*Figure 3K*). Based on these studies, we conclude that local proliferation and morphological changes, rather than infiltration of blood-borne cells, contribute to increased Iba1 immunoreactivity in axotomized DRGs.

## Construction of immune associated co-expression networks in axotomized DRGs

To gain insights into SNC-triggered genome wide transcriptional changes in DRGs, we carried out bulk RNA sequencing of ganglia harvested from naive, d1, d3, and d7 injured mice. To understand the modular network structure associated with peripheral axotomy, we carried out weighted gene co-expression network analysis (WGCNA) at different post-injury time points (*Geschwind and Konopka, 2009*; *Zhang and Horvath, 2005*). WGCNA permits identification of modules of highly co-expressed genes that likely function together. Focusing on prominently regulated gene modules, we find a previously described module (pink module [*Chandran et al., 2016*]), enriched for regeneration-associated gene (RAG) products, including *Jun, Fos, Stat3, Smad1, Atf3,* among other genes. In addition, WGCNA identifies a large turquoise module (*Figure 4A and B*), which along with the pink module, is stably upregulated following SNC (*Figure 4—figure supplement 1A and B*). To annotate module function, we applied gene ontology (GO) enrichment analyses, which showed enrichment (Benjamini-corrected p values < 0.05) for several GO categories associated with immune system function in the turquoise module. The enrichment plot for GO regulation shows a strong upregulation for *immune system processes* (*Figure 4C*). The most significantly upregulated GO terms include *cell activation*, *immune effector process*, and *defense response* (*Figure 4—figure supplement 1C*). Ingenuity pathway analysis (IPA) identified several upstream activators, including cytokines and growth factors (IFNγ, TNF, IL1b, IL6, TGFβ1, IL10, IL4, IFNβ1, IL2) and the transcription regulators STAT1, STAT3, IRF7, RELA (*Figure 4—figure supplement 1D*). The upregulation of immune system processes in axotomized DRGs correlates with a modest ~1.5-fold increase of gene products encoding the canonical macrophage markers *Itgam* (CD11b), *Aif1* (Iba1), and *Adgre1* (F4/80) (*Figure 4D–F*). In comparison, expression levels and fold-upregulation of *Atf3, Jun*, and *Stat3* are very robust (*Figure 4G–I*). Expression of the chemokine receptor *Ccr2* and the receptor subunits for the GM-CSF receptor (*Csf2ra* and *Csf2rb*) are elevated in axotomized DRGs; however, expression levels are low, especially for *Csf2rb* (~1 fpkm) (*Figure 4J–L*). Moreover, some of the immune gene activity observed in axotomized DRGs may involve non-hematopoietic cells. Collectively, RNA-seq provides independent evidence that SNC triggers a remote inflammatory response in DRGs, however this does not result in a massive increase in transcripts encoding canonical Mac markers. This conclusion is consistent with flow cytometry (*Figure 2B–G*), Western blot analysis (*Figure 2H and I*), and 3D reconstruction of Mac (*Figure 3G–K*) in axotomized DRGs.

## The cellular landscape of injured peripheral nerve tissue

To de-convolute the cellular complexity of injured sciatic nerve tissue in an unbiased manner, we applied scRNA-seq to capture the transcriptional landscape at single-cell resolution. Because injury-induced expansion of the immune compartment peaks around d3 (*Figure 1B*), we chose this time point to dissect and process whole nerves for single cell capture, using the 10x Genomics platform. A total of 17,384 cells was sequenced with 16,204 used for downstream analysis after removing cells with fewer than 200 genes, more than 7500, or mitochondrial content greater than 25%. Median unique genes per cell was 2507. More than 20 different cell clusters were identified using shared nearest neighbor clustering algorithm. Results are visualized using Uniform Manifold Approximation and Projection (UMAP) for dimension reduction (*Figure 5A*). The top 100 genes enriched in each cluster (*Figure 5—source data 1*) were used to assign cluster-specific cell identities. Most prominently featured are immune cells, identified by their strong expression of *Ptprc* (encoding CD45). Innate immune cells (*Itgam*/CD11b) make up a median 42.22% (±1.39%), and lymphocytes less than 1.73%(±. 27%), of the cells in the injured nerve (*Figure 5B and C*). Other abundantly featured cell types include mesenchymal progenitor cells (MES). We identify three distinct MES subpopulations (*Figure 5A*), reminiscent of a recent study examining the nerve response to digit tip amputation (*Carr et al., 2019*). In the injured sciatic nerve, MES make up 18.49% (±0.98%) of cells and differentially express the markers *Pdgfra* and *Sox9* (*Figure 5D and E*). MES are a rich source of extracellular matrix (ECM) molecules, including collagens (*Col1a, Col3a, Col5a, Col6a*), *Fn1*/fibronectin, *Fbn1*/fibrillin-1, *Lamb2*/laminin-b2, and numerous proteoglycans (*Figure 5—figure supplement 1A*). Individual MES clusters are identified as perineural MES (pMES) (*Slc2a1*/Glut1, *Itgb4*/integrin-β4, *Stra6*/stimulated by retinoic acid 6, *Sfrp5*/secreted frizzled related protein 5), endoneurial MES (eMES) (*Wif1*/Wnt inhibitory factor 1, *Bmp7*), and differentiating MES (dMES) (*Gas1*/Growth arrest-specific

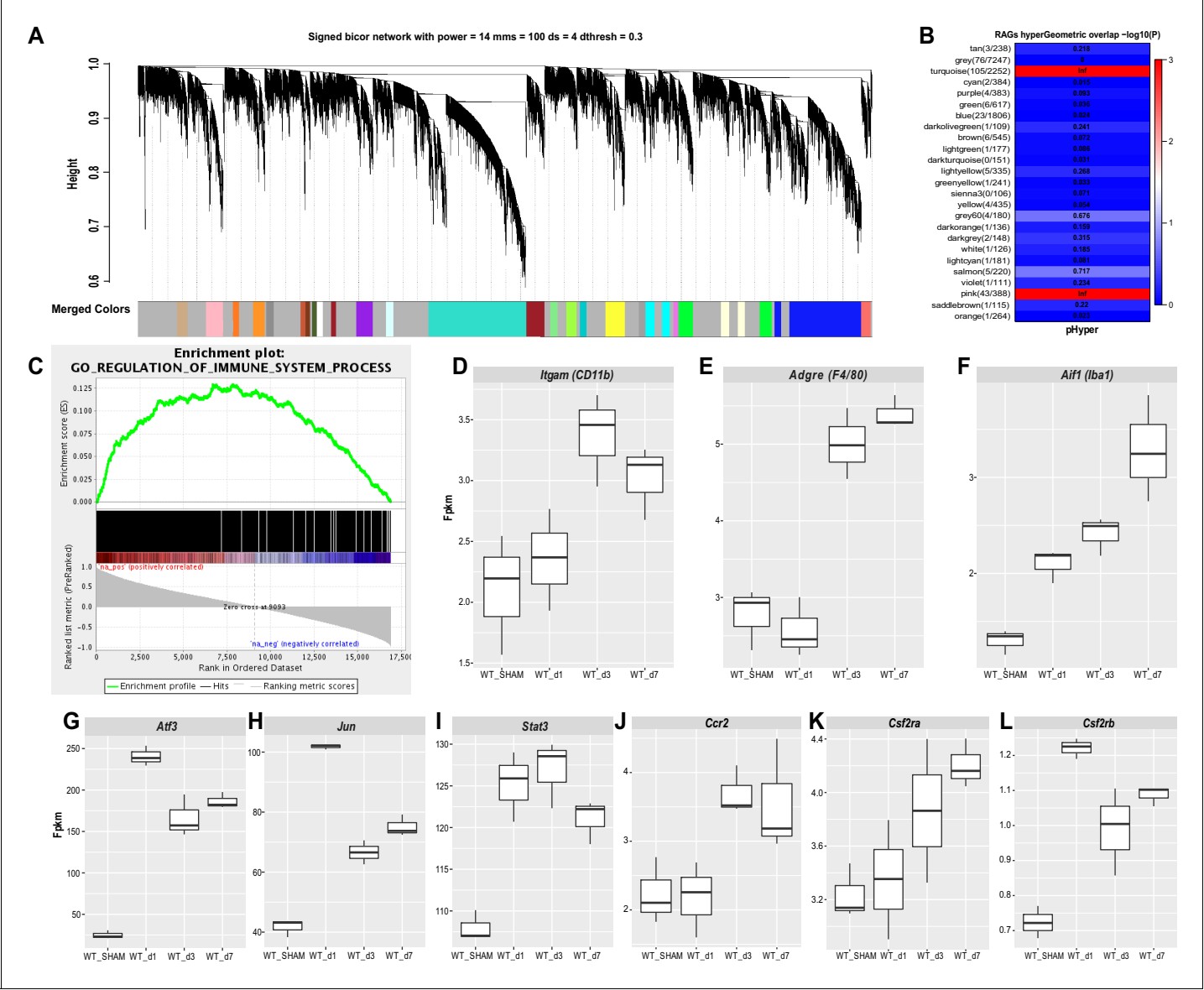

**Figure 4.** Stable upregulation of immune function associated gene co-expression networks in axotomized DRGs. Analysis of bulk RNAseq data from naïve and axotomized DRGs. DRGs were harvested from sham operated mice, d1, d3, and d7 post-SNC. (**A**) Network analysis of whole transcriptomes from naive and axotomized DRGs. Gene dendrogram identifies several co-expression modules. (**B**) Gene ontology (GO) analysis revealed significant and stable upregulation of the pink and turquoise modules. (**C**) Gene set enrichment analysis. Shown is the enrichment plot for GO terms of the turquoise module with overrepresentation of immune system processes. (**D–F**) Quantification of SNC-induced upregulation of commonly used macrophage markers *Itgam* (CD11b), *Adgre1* (F4/80), and *Aif1* (Iba1) in axotomized DRGs. (**G–I**) Quantification of SNC-induced upregulation of the RAGs *Atf3, Jun,* and *Stat3* in DRGs. (**J–L**) Quantification of SNC-induced upregulation of the chemokine receptor *Ccr2*, and the GM-CSF receptor subunits *Csf2ra* and *Csf2rb* in DRGs. Gene expression levels are shown as Fpkm (fragments per kilobase of transcript per million mapped reads). The online version of this article includes the following figure supplement(s) for figure 4:

**Figure supplement 1.** Module – trait relationships in axotomized DRGs and analysis of the turquoise module.

1, *Ly6a*/SCA-1, *Tnc/*tenascin, *Sfrp1*/secreted frizzled-related protein 1). The dMES cluster is fused to a small population of fibroblasts (Fb) (*Figure 5A*). STRING Reactome pathway analysis for MES clusters identifies *extracellular matrix organization* as top hit (*Figure 5—figure supplement 2*). Further analysis revealed that cells in eMES, but not in clusters pMES and dMES, are neural crest derived (*Carr et al., 2019*; *Gugala et al., 2018*).

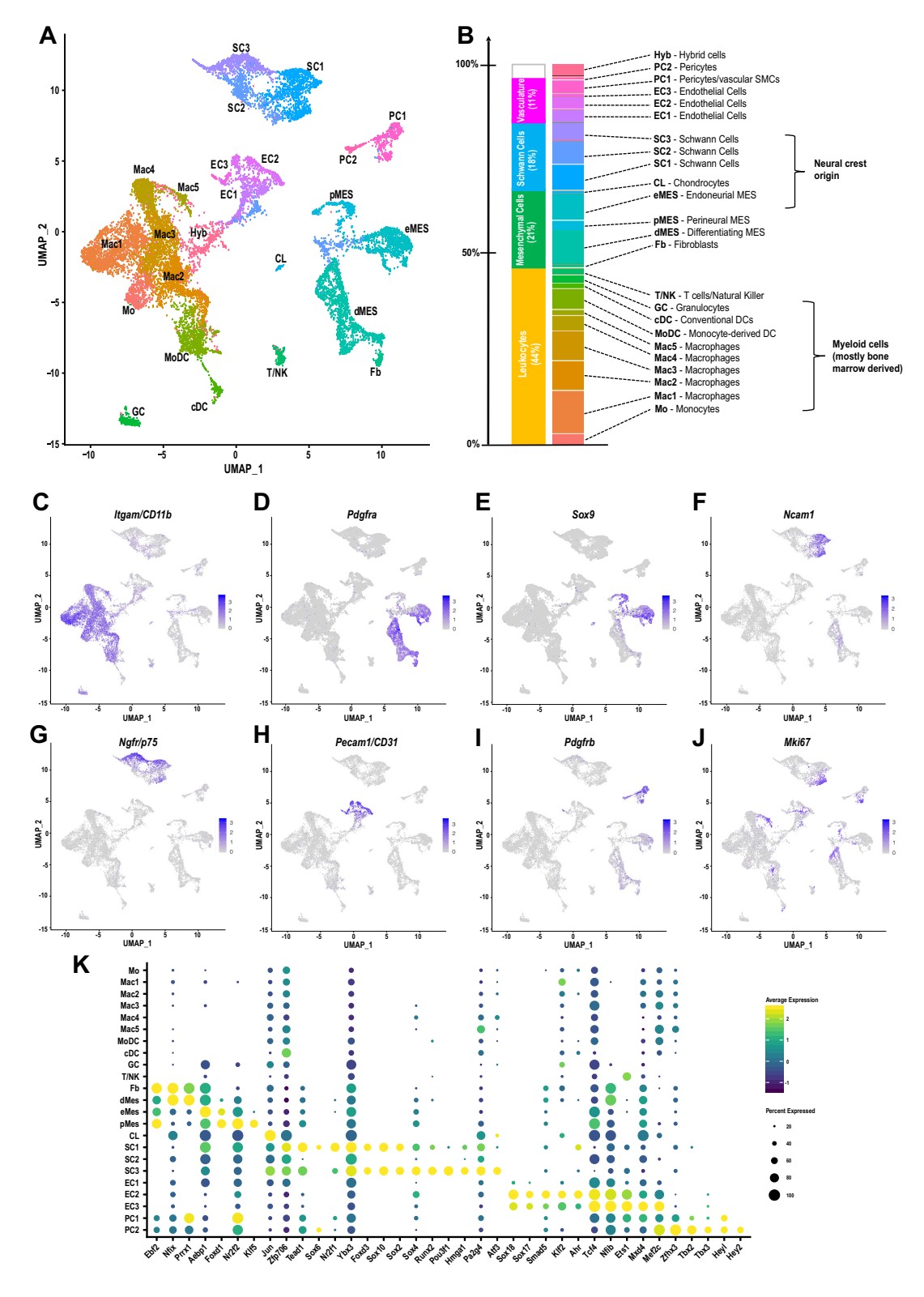

**Figure 5.** The cellular landscape of injured peripheral nerve. (A) Singe-cell transcriptome of injured mouse sciatic nerve at d3 post-SNC, n = 3 biological replicates. Unsupervised Seurat-based clustering identifies 24 cell clusters. Cell type identity for each cluster was determined by expression analysis of established markers. (B) List of all cell types identified by scRNA-sequencing. The size (percentile) of cell clusters and lineage relationships are shown. Abbreviations for cell cluster identities are indicated and used throughout the manuscript. (C–J) Feature plots of established cell markers

*Figure 5 continued on next page*

*Figure 5 continued*

used for identification of major cell types in the injured nerve. Shown are UMAP plots with markers for myeloid cells (*Itgam*/CD11b), fibroblast-like/mesenchymal cells (*Pdgfra, Sox9*), repair Schwann cells (*Ncam1, Ngfr*/p75), endothelial cells (*Pecam*/CD31), pericytes/smooth muscle vascular cells (*Pdgfrb*), and mitotically active cells (*Mki67*/Ki67). Expression levels are color coded and calibrated to average gene expression. (K) Dotplot shows cell-type-specific expression of the most abundant transcription regulators (TRs) in Fb, dMES, eMES, pMES, CL, SC1-3, EC1-3, PC1, and PC2 clusters identified by scRNA-seq of 3d injured sciatic nerve. Dotplot analysis shows the average gene expression (color coded) and percent of cells (dot size) that express the listed TRs in each cluster.

The online version of this article includes the following source data and figure supplement(s) for figure 5:

**Source data 1.** List of top 100 cluster enriched genes for all cell clusters identified in the 3d post-SNC nerve.
**Figure supplement 1.** Cell-cluster-specific expression of ECM components and molecules that regulate axon growth in the injured sciatic nerve.
**Figure supplement 2.** Single-cell gene expression in mesenchymal cell clusters in injured sciatic nerve.
**Figure supplement 3.** Single-cell gene expression in Schwann cell clusters in injured sciatic nerve.
**Figure supplement 4.** Single-cell gene expression in endothelial cell clusters in injured sciatic nerve.
**Figure supplement 5.** Single-cell gene expression in pericyte cell clusters in injured sciatic nerve.
**Figure supplement 6.** Single-cell gene expression of immune modulatory molecules in the injured PNS tissue.

Three clusters of Schwann cells (SC1-3) represent 17.48% (±1.53%) of cells in the injured nerve (*Figure 5A*). Cluster SC1 contains proliferating cells marked by *Mki67*/Ki67 expression (*Figure 5J*) and many cells that strongly express *Ncam1*, *Chl1*/cell adhesion molecule L1-like, *Erbb3*, *Epha5*, *Thbs2*/thrombospondin-2, *Tnc*, *Hbegf*, and the BMP antagonist *Sostdc1* (*Figure 5F* and *Figure 5—figure supplement 3*, *Figure 5—source data 1*). SC1 enriched transcription regulators (SC1-TR) include *Zfp706, Tead1, Sox6, Nr2f1*/COUP-TF (*Figure 5K*). SC3 cells express high levels of *Ngfr*/p75, *Nrcam*, *Gfra1*/GDNF family receptor alpha 1, *Btc*/betacellulin, *Gjb1*/connexin-32, *Cryab*/crystallin alpha B, *Tnfrsf12a*/Fn14, *Gadd45b* (*Figure 5G* and *Figure 5—figure supplement 3*, *Figure 5—source data 1*). SC3-TR include *Sox4, Runx2, Hmga1, Jun,* and the POU family member *Pou3f1*, a repressor of BMP and Wnt signaling, associated with a pro-myelinating cell state (*Figure 5K*). Cluster SC2, flanked by SC1 and SC3, expresses *nes*/nestin and *Cryab*. UMAP splits the SC2 cluster and places a subset of cells near MES cells, likely because of relatively higher expression in ECM encoding genes (*Bgn, Dcn,* and *Fn1*) compared to clusters SC1 and SC3. SC2 cells have a median 584 (±22) genes per cell and may have a higher degree of technical variation. STRING identified *axon guidance* and *integrin cell surface interactions* as top REACTOME pathways for SC1. *Axon guidance*, *gap junction assembly*, and *microtubule-dependent trafficking* are top hits for SC3 (*Figure 5—figure supplement 3*).

Cells associated with the nerve vasculature make up 14.2% (±3.19%). They include three clusters of endothelial cells (EC1-3), strongly expressing *Pecam*/CD31, representing 9.92% (±2.69%) of cells (*Figure 5H* and *Figure 5—figure supplement 4*). There are two pericyte cell clusters (PC1 and PC2) enriched for the pericyte markers (*Pdgfrb, Rgs5*) and vasculature-associated smooth muscle cells (*Acta2, Des, Myl9, Mylk),* representing 4.2% (±. 44%) (*Figure 5I* and *Figure 5—figure supplement 5*). A small cluster of chondrocyte-like cells (CL: *Comp*/cartilage oligomeric matrix protein, *Col27a1, Jun*) represents 0.5% (±0.44%). A cell cluster (3.09% (±1.08%)), designated Hyb, harbors few erythrocytes (*Hba, Hbb*) and some cell hybrids (Hyb). These cells had a median 521 (±27) expressed genes which was the lowest of any cell cluster and no clear identity could be assigned (*Figure 5B*).

Of relevance for neuronal regeneration, ECM components and numerous extracellular molecules known to regulate axon growth and regeneration are expressed by different cell types in the injured nerve (*Figure 5—figure supplement 1A*). MES and Fb are rich sources of gene products with neurotrophic and neurotropic properties, and thus may act in a paracrine fashion to regulate neuronal survival and direct axonal growth (*Figure 5—figure supplement 1B*). dMES express (*Igf1, Ogn*/osteoglycin, *Nid1*/Nidogen-1, *Ntn1*/netrin-1, *Postn*/periostin, *Gdf10*/BMP3b, *Cxcl12*/SDF1, *Dcn*/decorin, *Grn*/progranulin, *Sparc*/osteonectin, *Lamb2*/laminin-b2, *Serpinf1*), eMES (*Spp1, Dcn, Nid1*/nidogen-1, *Sparc, Serpine2*/glia-derived nexin, *Lum*/lumican, *Gpc3*/glycpican-3), and pMES (*Ntn1, Cldn1*/claudin-1, *Efnb2*/ephrin-b2, *Mdk*/midkine, *Nid1, Sdc4*/syndecan-4, *Thbs4*/thrombospondin-4, *Gpc3*). Repair Schwann cells in clusters SC1 and SC3 express high levels of cytokine receptor like factor 1 (*Crlf1*), and SC3 highly express cardiotrophin-like cytokine factor 1 (*Clcf1*). *Crlf1* and *Clcf1* are both members of the CNTF ligand family that signal through gp130. In addition, SC1 (*Chl1, Ncam1, Nrn1*/neuritin-1, *Ptn*/pleiotrophin, *Sema3e, Sema7a, Reln*/reelin), and SC3 (*Reln, Dag1*/dystroglycan,

*Gdnf*, *Nrcam*, *Sema3b*) express numerous membrane-bound and soluble factors with known roles in axon growth and guidance. Subpopulations of myeloid cells exhibit high expression of the osteo-pontin-encoding gene, *Spp1* and progranulin (*Grn*), powerful neurite outgrowth promoting factors (*Figure 5—figure supplement 1B*; *Altmann et al., 2016*; *Wright et al., 2014*). Taken together, scRNA-seq of injured nerve reveals that multiple cell types contribute to a large repertoire of extra-cellular molecules with neurotrophic and axon growth promoting properties.

## Mesenchymal progenitor cells in the injured nerve shape the inflammatory milieu

Non-hematopoietic cells in the injured nerve, including structural cells such as MES and Fb, show high immune gene activity and likely play a major role in shaping the inflammatory milieu (*Figure 5—figure supplement 6*). In comparison, repair SC exhibit low immune gene activity, suggesting they play a less important role in shaping nerve inflammation (*Figure 5—figure supplement 6*). In the 3d injured nerve, eMES express several chemokines (*Ccl2*, *Ccl7*, *Ccl9*, *Ccl11*/Eotaxin), *Mif*/Macrophage migration inhibitory factor, *Spp1*, *Thbs4*/Thrombospondin-4, and *Il33*. Cells in dMES express *Mif*, *Csf1*, *Cxcl14* and the complement components *C1s1*, *C1ra*, *C3*, *C4b*. Cells in pMES express *Ccl11*, *Cfh*/Complement factor h, *Mdk*, and *Thbs4*. Moreover, MES in the injured nerve likely contribute to wound healing and fibrosis, since they express several WNT pathway antagonists, including *Wfi1*, *Sfrp1*/Secreted frizzled related protein 1, *Sfrp2*, *Sfrp4*, and *Sfrp5* (*Figure 5—figure supplement 2* and *Figure 5—source data 1*). In the injured heart for example, blocking of WNT signaling was found to be critical to limit fibrosis and to promote differentiation of Mo into Mac (*Meyer et al., 2017*).

## The immune repertoire of injured sciatic nerve

The mononuclear phagocyte system (MPS) is comprised of Mo, Mac, and DC, cell types that are readily detected in the injured nerve by flow cytometry (*Figure 1*). UMAP, overlaid with Seurat-based clustering of scRNA-seq datasets, identified a connected continuum of seven cell clusters in the MPS (Mo, Mac1-5, and MoDC), characterized by strong expression of *Itgam*/CD11b (*Figure 5C*) and various degrees of the commonly used myeloid cell markers *Adgre1*/F4/80, *Aif1*/Iba1, *Cd68*, *Cx3cr1* and *Cd209a*/DC-SIGN (*Figure 6A–E*). Cells in the MPS strongly express the myeloid lineage-defining transcription factor PU.1 (*Spi1*). The C/EBP family member TR (*Cebpb*) is expressed by Mo/Mac, but not dendritic cells (*Figure 6F*). Myeloid cells are a rich source of fibronectin, extracellular proteases, and hydrolases (*Fn1*, *Tgfbi*, *Adam15*, *CtsC*, *CtsS*, *Gusb*) and likely play a major role in ECM remodeling, cell adhesion, and fibrosis. Monocytes strongly express *Ly6c2*/Ly6C, *Chil3*/chitinase-like 3, *Ifitm6*/interferon-induced transmembrane protein 6, *Itgal*/integrin αL, *Gsr*/glutathione reductase, *Hp*/haptoglobin (*Figure 6—figure supplement 1*). In addition, they express the TRs *Hif1a*, *Trps1*, and *Cebpb*/C-EBPβ, a bZIP TR important for Mo survival (*Figure 6F*). In the UMAP plot, the Mo cluster is flanked by three macrophage subpopulations (Mac1-Mac3) (*Figure 5A*). Mac1 cells express (*Fcgr2b*/Fc gamma receptor 2b, *Arg1*/arginase-1, *Ltc4s*/leukotriene C4 synthase, *Lpl*/lipoprotein lipase, *Camkk2*). Mac2 (*Cx3cr1*, *Ccr2*, *Csf1r*) and Mac3 (*Cx3cr1*, *Mrc1*/CD206, *Ccr2*, *Adgre1*/F4/80, *Csf1r*, *Cd38*) express overlapping, yet distinct, sets of surface receptors (*Figure 6—figure supplement 1*). Of note, individual Mac subpopulations often co-express markers traditionally associated with M1-like and M2-like cells, indicating that these markers are of limited use to describe the more complex physiological states of Mac subpopulations in the injured nerve. Mac4 cells are characterized by high levels of *Trem2*/ triggering receptor expressed on myeloid cells 2, *Arg1*/arginase-1, *Pf4*/CXCL4, *Stab1*/stabilin-1, *Cd68* (*Figure 6—figure supplement 1*) and express the TRs *Cebpa*, *Mafb*, *Mef2a* (*Figure 6K*). Cluster Mac5 is small, 239 cells, and harbors dividing (*Mki67*) myeloid cells with 'stem-like' features (*Stmn1*/ Stathmin-1, *Top2a*, *Hmgb2*, *Tupp5*) (*Figure 5J*, *Figure 6—figure supplement 1*, *Figure 5—source data 1*). In addition, a smaller group of dividing cells (*Mki67*, *Top2a*) is embedded in the MPS and located between clusters Mac2 and MoDC (*Figure 6—figure supplement 2A and B*). To distinguish between dividing nerve resident myeloid cells and dividing blood-derived myeloid cells, we subjected WT-tdTom parabionts to SNC (*Figure 3A*). At 3d post-SNC, WT nerves were analyzed for tdTom$^+$ cells that co-stain with anti-Ki67 and anti-F4/80 (*Figure 6—figure supplement 2C*). TdTom$^+$F4/80$^+$Ki67$^+$ cells were identified, indicating that blood-borne, stem-like myeloid cells are present in the injured sciatic nerve. Mac2 cells express high levels

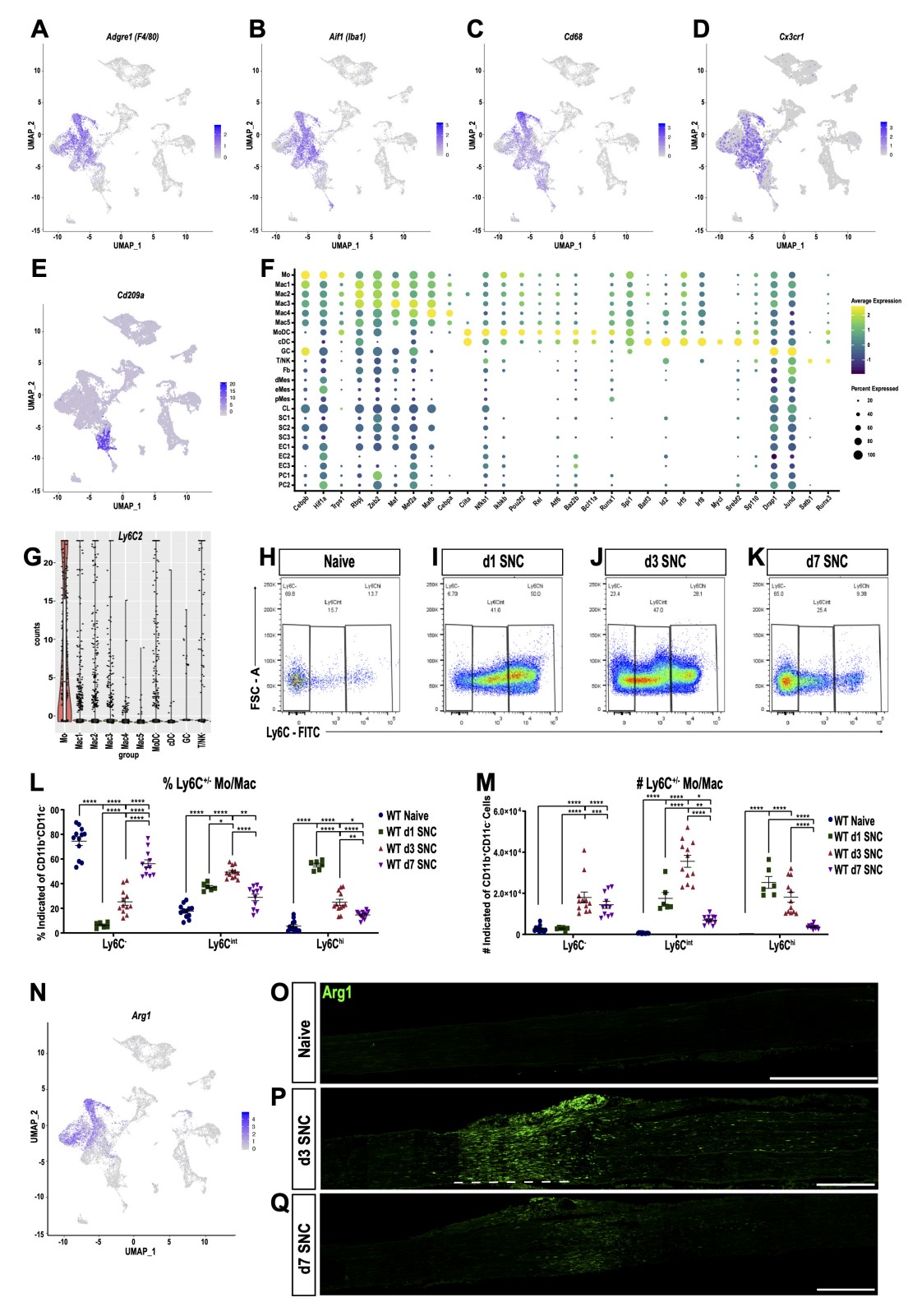

**Figure 6.** Macrophage subpopulation in the injured nerve are functionally distinct and localize to specific sites. (A–E) Feature plots of *Adgre1* (F4/80), *Aif1* (Iba1), *Cd68* (Scavenger receptor class D), *Cx3cr1* (Fractalkine receptor), and *CD209a* (DC-SIGN) expression in the d3 post-SNC nerve. (F) scRNAseq dot plot analysis of transcription regulators (TRs) enriched in leukocytes. Average gene expression and percentage of cells expressing the TR are shown. (G) Violin plot of *Ly6c2* (Ly6C) expression in immune cells of the d3 post-SNC nerve. (H–K) Flow cytometric analysis of sciatic nerve Mo/Mac

*Figure 6 continued on next page*

*Figure 6 continued*

(CD45[+], CD11b[+], Ly6G[-], CD11c[-]) in naive mice, d1, d3, and d7 post-SNC mice. Mo/Mac maturation was assessed by Ly6C surface staining. (**L, M**) Quantification of Ly6C distribution on Mo/Mac in naïve nerves and at different post-SNC time points (n = 11 biological replicates per time point); (**L**) Percentile of Ly6C[-], Ly6C[int], and Ly6C[hi] Mo/Mac and (**M**) number of Ly6C[-], Ly6C[int], and Ly6C[hi] Mo/Mac. Flow data are represented as mean ± SEM. Statistical analysis was performed in GraphPad Prism (v7) using one-way or two-way ANOVA with correction for multiple comparisons with Tukey's post-hoc test. A p value < 0.05 (*) was considered significant. p<0.01 (**), p<0.001 (***), and p<0.0001 (****). (**N**) Feature plot showing *Arg1* (Arginase-1) expression in the 3d post-SNC nerve. (**O–Q**) Longitudinal sciatic nerve sections of *Arg1-YFP* reporter mice, from naive mice (**O**), d3 (**P**), and d7 (**Q**) post-SNC mice. YFP[+] cells (green) are localized to the injury site (underlined with a dashed line), proximal is to the left. Representative example of n = 4 biological replicates. Scale bar, 200 µm.

The online version of this article includes the following figure supplement(s) for figure 6:

**Figure supplement 1.** Single-cell gene expression in myeloid cell clusters of injured sciatic nerve.

**Figure supplement 2.** Identification of blood-borne, stem-like myeloid cells in the injured sciatic nerve.

**Figure supplement 3.** Monocyte-to-macrophage differentiation based on single-cell expression modeling.

**Figure supplement 4.** Mo/Mac maturation in axotomized DRGs, assessed by Ly6C surface expression.

**Figure supplement 5.** Evidence for distinct immune compartments in the injured sciatic nerve.

**Figure supplement 6.** Expression of gene products that regulate cholesterol transport and metabolism in macrophages of injured and naive nerve.

of MHCII genes (*H2-Aa, H2-Ab1, H2-Eb1, M2-DM*) and the CD74 invariant chain of MHCII (*Cd74*), typically associated with antigen presentation to CD4[+] T cells. The MPS harbors monocyte-derived dendritic cells (MoDC), professional antigen-presenting cells, characterized by high level expression of MHCII genes, *Itgax/*CD11c, *Itgb7/*integrin-β7, *Napsa/*Napsin-A, and *Cd209a/*DC-SIGN (*Figure 6E*, *Figure 6—figure supplement 1*). Mac2 and MoDC express *Ciita* (*Figure 6F*), a class II transactivator, that promotes MHCII gene expression (*Accolla et al., 2019*). Few plasmacytoid DCs (pDC) (*Siglech, Ly6d*) and conventional DCs (cDC) (*Clec9a, Xcr1, Itgae, Tlr3, Ifi205, Cd24a, Btla/* CD272) are detected in the MPS (*Figure 6—figure supplement 2D and E*). cDC show enriched expression of the TRs *Batf3, Id2, Irf5, Irf8, Mycl, Srebf2* (*Figure 6F*). DC clusters can readily be distinguished from other myeloid cells, based on their expression of *Bcl11a*, a TR that determines DC fate (*Ippolito et al., 2014*). Cells in the MoDC cluster show high expression of the TRs *Nfkb1, Pou2f2, Runx1, Rel/*c-Rel, and *Ikbkb/*IKKβ (*Figure 6F*). The GC cluster in the d3 nerve is small, 314 cells, and mainly includes neutrophils (*S100a8, S100a9, Mmp9, Retnlg/*Resistin-like gamma), intermingled with few eosinophils (*Siglecf*) (*Figure 5A*, *Figure 6—figure supplement 1*). Overall, the Seurat cluster analysis is in good agreement with the abundance and identity of immune cell profiles detected by flow cytometry and also reveals the presence of a large and connected continuum of cell states in the myeloid compartment (*Figure 5A*). To infer the most probable differentiation trajectories from Mo toward their descendants, we used Slingshot, a method for pseudo-time trajectory analysis (*Street et al., 2018*). The analysis reveals a bifurcated trajectory and provides independent evidence that blood-borne Mo entering the nerve give rise to different Mac subpopulations as well as MoDC. The predicted differentiation trajectory indicates that Mo first give rise to Mac3, and cells in cluster Mac3 then differentiate either into Mac1, Mac2, or Mac4 cells. Furthermore, Mac2 cells are predicted to differentiate into MoDC (*Figure 6—figure supplement 3*).

The 'connected continuum' of Mo/Mac in the injured nerve, as revealed by scRNA-seq, was independently verified by flow cytometry. The Mo/Mac population (CD45[+]CD11b[+]Ly6G[-]CD11c[-]) is highly plastic and can be subdivided based on surface levels of the lymphocyte antigen 6C (Ly6C). Ly6C is expressed at high levels on proinflammatory, circulating monocytes and is downregulated as they infiltrate tissues and mature into macrophages and dendritic cells (*King et al., 2009*). As expected, scRNA-seq of injured nerve shows that *Ly6c2*, the gene encoding Ly6C, is strongly expressed by Mo, but much less by Mac subpopulations (*Figure 6G*). Flow cytometry shows that naive nerve tissue harbors a small Mac population, mostly comprised of Ly6C[-] (70%) cells and few Ly6C[int] (16%) and Ly6C[hi] (14%) cells (*Figure 6H*). At d1 post-SNC, the number of Mo/Mac increases sharply and Ly6C distribution is skewed toward classically activated Ly6C[hi] cells (50%), with fewer Ly6C[int] (41%) and Ly6C[-] (9%) cells (*Figure 6I*). At d3, Ly6C[hi] (28%), Ly6C[int] (47%), and Ly6C[-] (25%) cells are detected (*Figure 6J*) and at d7, the majority of Mo/Mac are non-classical Ly6C[-] (65%) and intermediate Ly6C[int] (25%), with few Ly6C[hi] cells (10%) (*Figure 6K*). This shows that Ly6C[hi] Mo migrate into the injured nerve in large numbers and increase inflammation during the acute phase. Later, as nerve inflammation resolves, the Mo/Mac number and polarization gradually return back to pre-injury homeostatic

levels (*Figure 6L and M*). Noteworthy, the Mo/Mac population in axotomized DRGs shows an opposite response with regard to surface Ly6C distribution. In naive DRGs, Mo/Mac are comprised of Ly6C⁻ (30%), Ly6C$^{int}$ (27%), and Ly6C$^{hi}$ (43%) cells. Upon SNC, the distribution shifts to 75%, 16%, and 9% on d1, to 53%, 20%, and 27% on d3, and 52%, 23%, and 25% on d7 (*Figure 6—figure supplement 4*). Together, these data show that SNC triggered inflammation in the nerve is massive and characterized by a short pro-inflammatory phase that rapidly transitions to a resolving state. A similar immune response is not observed in axotomized DRGs.

## Identification of macrophage subpopulations with distinct functions and distribution patterns in the injured nerve

Mac subpopulations show overlapping, yet distinct, expression patterns of the canonical markers *Adgre1*(F4/80), *Aif1*(Iba1), *Cd68,* and *Cx3cr1* (*Figure 6A–D*). Moreover, cells in Mac4 and some cells in clusters Mac1 and Mac3 express high levels of *Arg1*, while other Mac subpopulations do not (*Figure 6N*). To explore tissue distribution of *Arg1*⁺ cells relative to F4/80⁺ and CD68⁺ cells in naive and injured nerves, we subjected *Arg1-YFP* reporter mice to SNC. In naive mice, no YFP⁺ cells are observed (*Figure 6O*) while few F4/80⁺ and CD68⁺ are detected (*Figure 6—figure supplement 5*). At d1, few YFP⁺ cells accumulate near the injury site (data not shown) and at d3 many more are present (*Figure 6P*). Unexpectedly, YFP⁺ cells are confined to the nerve crush site and largely absent from the distal nerve stump. This stands in contrast to F4/80⁺ and CD68⁺ macrophages, found at the injury site and the distal nerve (*Figure 6—figure supplement 5*). At d7, only few *Arg1-YFP*⁺ cells are found at the injury site and none in the distal nerve stump (*Figure 6Q*). F4/80⁺ Mac, on the other hand, are more uniformly distributed within the injury site and distal nerve stump (*Figure 6—figure supplement 5*). This shows the existence of different immune compartments in the injured nerve. A subpopulation of *Arg1*⁺ macrophages (including cells in cluster Mac4) is preferentially localized to the crush site, whereas F4/80⁺ macrophages (including cells in cluster Mac2 and Mac3) are abundant in the distal nerve where fibers undergo Wallerian degeneration. Pathway analysis of cell clusters in the innate immune compartment reveals common functions in phagocytosis, phagosome, and endolysosomal digestion, but also highlights important differences (*Figure 6—figure supplement 1*). KEGG pathways specific for Mo include *cytokine signaling* and *leukocyte trans-endothelial migration*, providing independent evidence for their hematogenous origin. Mo are highly plastic and predicted to give rise to monocyte-derived Mac subpopulations in the injured nerve (*Figure 6—figure supplement 3*). Top KEGG pathways for Mac3 are *chemokine signaling pathway*, *complement and coagulation cascades*, and *cytokine-cytokine receptor interaction* (*Figure 6—figure supplement 1*). For Mac1 cells, *complement and coagulation cascades*, suggesting that Mac1 and Mac3 play roles in opsonization and blocking of endoneurial bleeding. For Mac2 cells, KEGG pathway analysis identified *Leishmaniasis* and *Tuberculosis* as top hits (*Figure 6—figure supplement 1*). For Mac4 cells, pathway analysis identified *negative regulation of immune system processes* and *cholesterol metabolism*. Cholesterol metabolism in Mac4 cells includes gene products that regulate reverse *cholesterol transport* (*Abca1/*ATP binding cassette subfamily A1, *Abcg1/*ATP-binding cassette subfamily G1, *Ctsd/*Cathepsin-D, *Ctsb/*Cathepsin-B), *cholesterol and lipid storage* (*Plin2/*perilipin), *formation of cholesterol esters (Soat1), cholesterol ester hydrolysis and lipoprotein metabolism* (*Lipa/*lipase-A, *Nceh1/*Neutral cholesterol hydrolase 1, *Apoe/*Apolipoprotein E) and *intracellular cholesterol transport* (*Npc2/*Niemann-Pick C2 and *Scarb2/*Scavenger receptor class B member 2) (*Figure 6—figure supplement 6A–6I*). The abundance of gene products that protect from cholesterol overloading (*Haidar et al., 2006*; *Viaud et al., 2018*; *Wu et al., 2018*), suggests that this cluster is comprised of cholesterol laden cells. Importantly, tissue resident macrophages in naive nerves (*Wang et al., 2020*), either do not express cholesterol regulatory gene products, or express them at significantly lower levels (*Figure 6—figure supplement 6J–6S*).

## Cell-type-specific expression of engulfment receptors in the injured nerve

In the injured nerve, blood-borne phagocytes and repair SC collaborate in myelin removal. Repair SC use the receptor tyrosine kinases AXL and MerTK for myelin phagocytosis (*Brosius Lutz et al., 2017*). Clusters SC1 (*Axl$^{hi}$, Mertk⁻*) and SC3 (*Axl$^{low}$, Mertk$^{int}$*) exhibit differential expression of these two receptors (*Figure 7—figure supplement 1*). Interestingly, *Axl* and *Mertk* expression in myeloid

cells is very low, suggesting that innate immune cells and repair SC employ different mechanisms for myelin phagocytosis. Mac subclusters strongly express the myelin-binding receptors *Lrp1* (low-density lipoprotein receptor-related protein 1), *Pirb* (paired Ig-like receptor B), *Cd300lf* (sphingomyelin receptor), and several scavenger receptors (*Msr1, Cd36, Cd68*), including high levels of opsonic receptors (*Fcgr1, Fcgr3, Fcgr4, Fcer1g*) that may contribute to phagocytosis of antibody marked myelin debris (*Figure 7—figure supplement 1* and *Figure 5—source data 1*; *Atwal et al., 2008*; *Grajchen et al., 2018*; *Izawa et al., 2014*; *Kuhlmann et al., 2002*; *Stiles et al., 2013*). Compared to Mo/Mac of injured nerves, phagocytosis receptor expression is much lower in naive nerve Mac (*Figure 7—figure supplement 1*).

In addition to debris phagocytosis, myeloid cells participate in removal of apoptotic cells (AC), primarily dying neutrophils and other leukocytes. Phagocytic uptake of AC, called efferocytosis, is mediated by a range of specialized engulfment receptors and mechanisms for ingestion (*Boada-Romero et al., 2020*). AC are selectively recognized due to phosphatidylserine (PS) or calreticulin (*Calr*) accumulation on their surface; both function as strong 'eat me' signals (*Figure 7A*). Conversely, healthy cells display the 'don't eat me' signal CD47 that binds to the cell surface receptor SIRPα (signal regulatory protein α) encoded by *Sirpa*, to block efferocytosis (*Kourtzelis et al., 2020*). *Calr* and *Cd47* are boadly expressed by cells in the injured nerve, while *Sirpa* is largely confied to myeloid cells (*Figure 7B*). PS is directly recognized by cell surface receptors such as CD300 family members (*Cd300a, Cd300lb, Cd300lf*), stabilin-1 (*Stab1*), and oxidized-PS by the scavenger receptor *Cd36*, molecules that are expressed by phagocytes in the injured nerve (*Figure 7C*). Alternatively, PS binds indirectly, via bridging molecules, to engulfment receptors (*Voss et al., 2015*). Interestingly, in the injured sciatic nerve, numerous cell types express specific sets of bridging molecules, indicating that they may contribute in an autocrine or paracrine manner to AC removal. Bridging molecules prominently expressed include complement C1q components (*C1qa, C1qb, C1qc, C1ra*), annexins (*Anxa1-5*), pentraxin (*Ptx3*), thrombospondin 1 (*Thbs1*), collectin kidney protein 1 (*Colec11*), soluble collectin placenta 1 (*Colec12*), galectin-3/MAC-2 (*Lgals3*), growth arrest-specific 6 (*Gas6*), protein S (*Pros1*), milk fat globule-EGF factor 8 (*Mfge8*), and apolipoprotein E (*Apoe*) (*Figure 7B*). Bridging molecules that bind to PS are recognized by a large and diverse set of engulfment receptors on phagocytes, including Lrp1, Trem2, Dap12 (*Tyrobp*), C1q receptor (*C1qr/Cd93*), C3a receptor 1 (*C3ar1*), integrin αMβ2, (*Itgam, Itgb2*), integrin αv (*Itgav*), integrin β3 (*Itgb3*), CD14, and members of the scavenger receptor family (*Cd68* and *Msr1*/Mac scavenger receptor 1) (*Doran et al., 2020*; *Erriah et al., 2019*; *Korns et al., 2011*). Strikingly, many of these engulfment receptors are expressed by myeloid cells and are particularly abundant in cluster Mac4 (*Figure 7C*). Indirect evidence that Mac4 cells eat AC corpses, is the strong expression of gene products that regulate lipid metabolism and mechanisms that protect cells from excessive cholesterol loading, such as reverse cholesterol transport and cholesterol esterification (*Figure 6—figure supplement 6*). To assess whether expression of gene products involved in efferocytosis are upregulated following nerve injury, we took advantage of recently published scRNA-seq data sets generated from naive mouse sciatic nerve tissue (*Wang et al., 2020*; *Ydens et al., 2020*). Importantly, bridging molecules and engulfment receptors are either not expressed by macrophages in the naive nerve, or expressed at much lower levels than in Mac4 cells in the injured nerve (*Figure 7—figure supplement 2*).

## Efferocytosis of leukocytes in the injured sciatic nerve

To directly test whether efferocytosis takes place in the injured nerve, we first examined the presence of AC corpses. Viability-dye labeling, combined with flow cytometry, identified an increase in AC at d3 and d7 post-SNC (*Figure 7D*). During nerve debridement, degenerated nerve fibers and AC corpses are removed. In order to distinguish between efferocytosis of dying leukocytes and phagocytosis of nerve fiber debris, we generated WT$^{CD45.1}$-tdTom$^{CD45.2}$ parabiotic mice (*Figure 7E*). Both mice in the parabiosis complex were subjected to bilateral SNC. At d3 post-SNC, live cells in the injured WT$^{CD45.1}$ nerve were analyzed by flow cytometry (gating strategy is illustrated in *Figure 3—figure supplement 1*). All tdTom$^+$ cells in the injured nerve of the WT$^{CD45.1}$ parabiont are blood-borne immune cells. Moreover, cells that are CD45.1$^+$tdTom$^+$CD45.2$^-$ represent tdTom$^+$ leukocytes that were eaten in the nerve by CD45.1$^+$ phagocytes. In non-parabiotic (single) tdTom mice,~95% of myeloid cells (CD11b$^+$) in the 3d injured nerve are tdTom$^+$ (*Figure 7F*) and in the WT$^{CD45.1}$ parabiont ~39% are CD11b$^+$tdTom$^+$ (*Figure 7G*). Importantly, in the WT$^{CD45.1}$ parabiont, CD45.1$^+$tdTom$^+$CD45.2$^-$ (Q3) cells are readily detected in the injured nerve and such cells are not

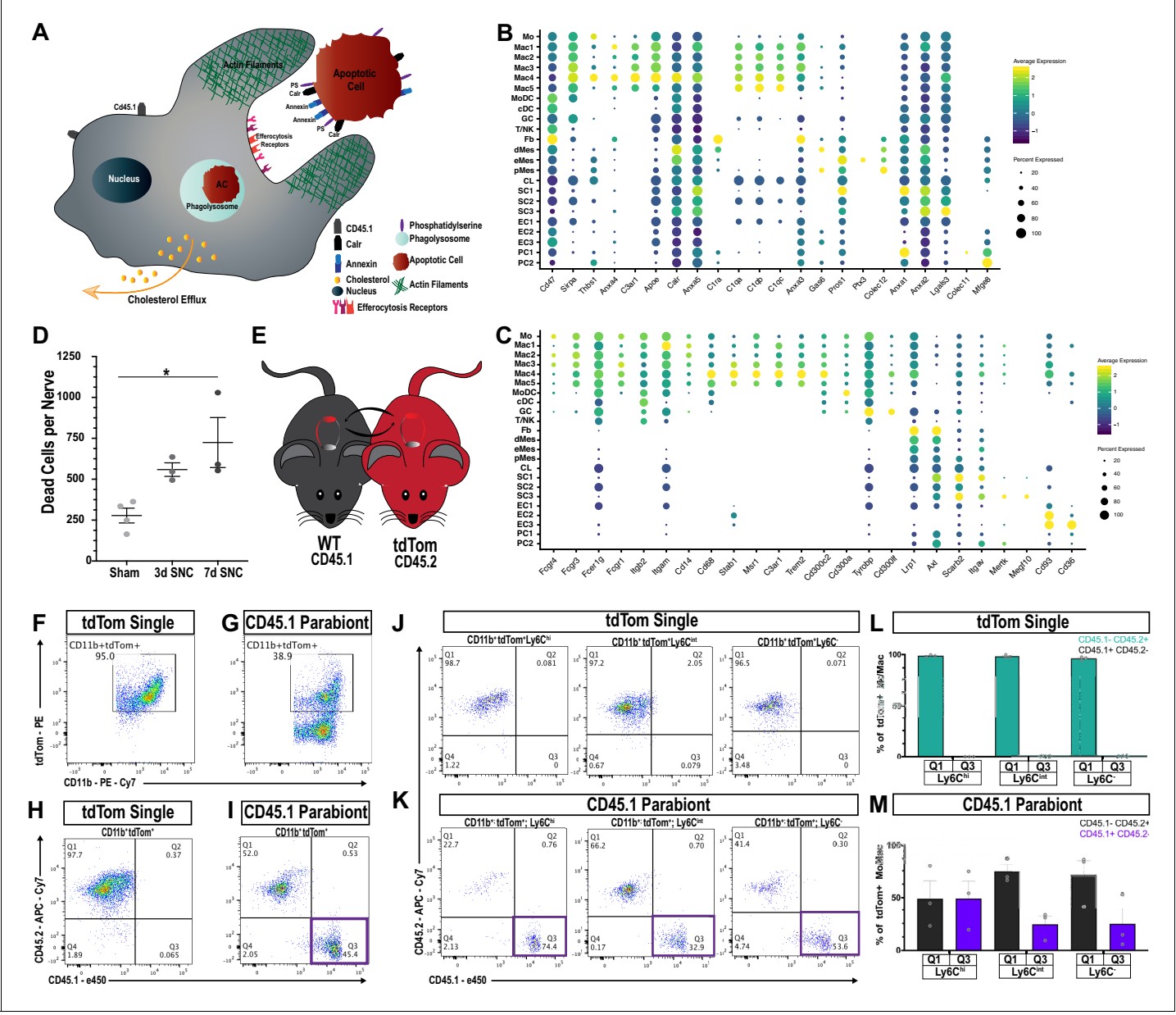

**Figure 7.** Macrophages 'eat' dying leukocytes in the injured nerve. (**A**) Cartoon of phagocyte with actin rich phagocytic cup eating a tdTom⁺ apoptotic cell (AC). 'Eat me' signals displayed on the surface of AC allow direct or indirect recognition via engulfment receptors. Following engulfment by phagocytes, AC are digested in the phagolysosome. Cellular cholesterol levels are controlled by upregulation of specific efflux mechanisms. (**B**) scRNAseq dotplot analysis of 'don't eat me' molecules (*Cd47, Sirpa*) and bridging molecules prominently expressed across cell types in the d3 post-SNC nerve. Average gene expression and percentage of cells expressing the gene are shown. (**C**) scRNAseq dotplot analysis of engulfment receptors in the d3 post-SNC nerve. Average gene expression and percentage of cells expressing the gene are shown. (**D**) Flow cytometric analysis of dead cells accumulating in the d3 and d7 nerve (n = 3 biological replicas per time point). Data are represented as mean ± SEM. (**E**) Parabiosis complex of WT (CD45.1) mouse with a (CD45.2) tdTom reporter mouse. (**F**) Flow cytometry dot plot showing tdTom⁺ myeloid cells (CD45.2⁺, CD11b⁺) in the sciatic nerve of non-parabiotic (tdTom single) mice. (**G**) Flow cytometry dot plot showing tdTom⁺ myeloid cells (CD11b⁺) in the sciatic nerve of the WT CD45.1 parabiont. (**H**) Flow cytometry dot plot of CD11b⁺, tdTom⁺-gated cells from non-parabiotic (tdTom⁺ single) mice, analyzed for CD45.1 and CD45.2 surface staining. (**I**). Flow cytometry dot plot of CD11b⁺, tdTom⁺-gated cells from the CD45.1 parabiont, analyzed for CD45.1 and CD45.2 surface staining. Quadrant 3 (Q3) identifies CD45.1⁺, tdTom⁺, CD45.2⁻ myeloid cells, indicative of ongoing efferocytosis. (**J**) Flow cytometry dot plots of Mo/Mac in the injured nerve of non-parabiotic (tdTom single) mice. Mo/Mac maturation was assessed by Ly6C surface staining. Shown are monocytes (Ly6Cʰⁱ), Mo/Mac (Ly6Cⁱⁿᵗ), and Mac (Ly6C⁻). (**K**) Flow cytometry dot plots of Mo/Mac in the injured nerve of the CD45.1 parabiont. Shown are monocytes (Ly6Cʰⁱ), Mo/Mac (Ly6Cⁱⁿᵗ), and Mac (Ly6C⁻). The quadrant with CD45.1⁺, tdTom⁺, CD45.2⁻ cells (Q3) is highlighted. Biological replicates n = 3, with three parabiotic pairs per replica. (**L, M**) Quantification of CD45.1⁺, tdTom⁺, CD45.2⁻ cells in quadrant Q3 and CD45.2⁺, tdTom⁺, CD45.1⁻

*Figure 7 continued on next page*

*Figure 7 continued*

cells in Q1. (**L**) In the injured nerve of (tdTom single) mice, no CD45.1$^+$ cells are detected. (**M**) In the injured nerve of the WT CD45.1 parabiont, CD45.1$^+$, tdTom$^+$, CD45.2$^-$ Mo (Ly6C$^{hi}$), Mo/Mac (Ly6C$^{int}$), and Mac (Ly6C$^-$) are found; n = 3 biological replicates, with three parabiosis pairs pooled per replicate.

The online version of this article includes the following figure supplement(s) for figure 7:

**Figure supplement 1.** Expression of gene products implicated in myelin binding and phagocytosis.
**Figure supplement 2.** Expression analysis of bridging molecules and engulfment receptors in macrophages of naive nerve and injured nerve.
**Figure supplement 3.** Contribution of MoDC to efferocytosis in the injured sciatic nerve.

present in tdTom (single) mice (*Figure 7H and I*). This indicates that efferocytosis of apoptotic leukocytes takes place in the injured nerve. To determine which immune cell types eat apoptotic leukocytes, we analyzed CD45.1$^+$tdTom$^+$ cells for surface levels of Ly6C and CD11c to distinguish between maturing Mo/Mac (Ly6C$^{hi}$ to Ly6C$^-$) and MoDC (CD11c$^+$). Mo/Mac have the biggest appetite for tdTom$^+$ apoptotic leukocytes, more so than MoDC, suggesting they remove the bulk of dying leukocytes (*Figure 7K and M* and *Figure 7—figure supplement 3*). As negative controls, non-parabiotic tdTom$^{CD45.2}$ mice were processed in parallel (*Figure 7J and L – Figure 7—figure supplement 3*). Collectively, these studies show that efferocytosis of dying leukocytes takes place in the injured sciatic nerve, and thus, serves as an important mechanism to clear the nerve of AC corpses.

## *Csf2* deficiency skews the immune response in the injured nerve toward classically activated Ly6C$^{hi}$ monocytes

While PNS injury elicited inflammation is important for axon regeneration, it is not clear whether inflammation in the nerve or axotomized DRGs is a primary driver of peripheral axon regeneration, or conditioning-lesion-induced central axon growth (*Figure 8A*). Bulk RNA-seq of axotomized DRGs and scRNA-seq of injured nerve identified chemokine and cytokine ligand-receptor systems preferentially expressed in the injured nerve. GM-CSF signaling is of interested because this cytokine is present in the injured nerve and has been implicated in neuroprotection and axon repair (*Be'eri et al., 1998*; *Franzen et al., 2004*; *Legacy et al., 2013*). Moreover, GM-CSF increases surface expression of galectin-3 (*Saada et al., 1996*) and in non-neural tissues galectin-3 functions as a bridging molecule for efferocytosis of apoptotic immune cells (*Erriah et al., 2019*; *Wright et al., 2017*). Transcripts for the GM-CSF receptor subunits (*Csf2ra* and *Csf2rb*) are abundantly expressed by myeloid cells in the injured nerve (*Figure 8B and C*), but not in axotomized DRGs (*Figure 4K and L*). To assess the role in nerve-injury-triggered inflammation, we employed *Csf2$^{-/-}$* mice (*Figure 8—figure supplement 1*) and subjected them to SNC. Flow cytometry was used to quantify immune cell profiles in naïve nerves and at 1d, 3d, and 7d post-SNC. In naïve WT and *Csf2$^{-/-}$* mice, the number of endoneurial Mac is comparable, and the majority of them are Ly6C$^-$ or Ly6C$^{int}$ cells (*Figure 8D and E*). In the d3 injured nerve, there is a strong increase in the Mo/Mac population, in both, WT and *Csf2$^{-/-}$* mice (*Figure 8F and G*). However, when analyzed for surface Ly6C expression, significantly fewer Ly6C$^-$ cells are present in *Csf2$^{-/-}$* mice. Conversely, the population of Ly6C$^{hi}$ cells is significantly elevated in *Csf2$^{-/-}$* mice when compared to WT mice (*Figure 8H*). This indicates that Mo/Mac maturation and inflammation resolution in the injured nerve of *Csf2$^{-/-}$* mice is significantly delayed. Delayed maturation is only observed in the Mo/Mac population, since analysis of surface Ly6C expression on MoDC is comparable between WT and *Csf2$^{-/-}$* mice (*Figure 8I*).

## *Csf2* is required for conditioning-lesion-induced dorsal column axon regeneration

To assess whether proper Mo/Mac maturation in the injured nerve is important for conditioning-lesion-induced regeneration of central axon projections, adult WT and *Csf2$^{-/-}$* mice were either subjected to bilateral SNC or sham operated. Seven days later, a dorsal column lesion (DCL) was placed at cervical level 4 of the spinal cord. Five weeks following DCL, cholera toxin B (CTB) traced dorsal column axons were analyzed in longitudinal spinal cord sections (*Figure 8A*). DCL causes axon 'dieback' (*Horn et al., 2008*). In WT mice without conditioning lesion, there is a 600 ± 80 μm gap between the lesion center, and the most proximal, CTB labeled axons (*Figure 8J and K*). In WT

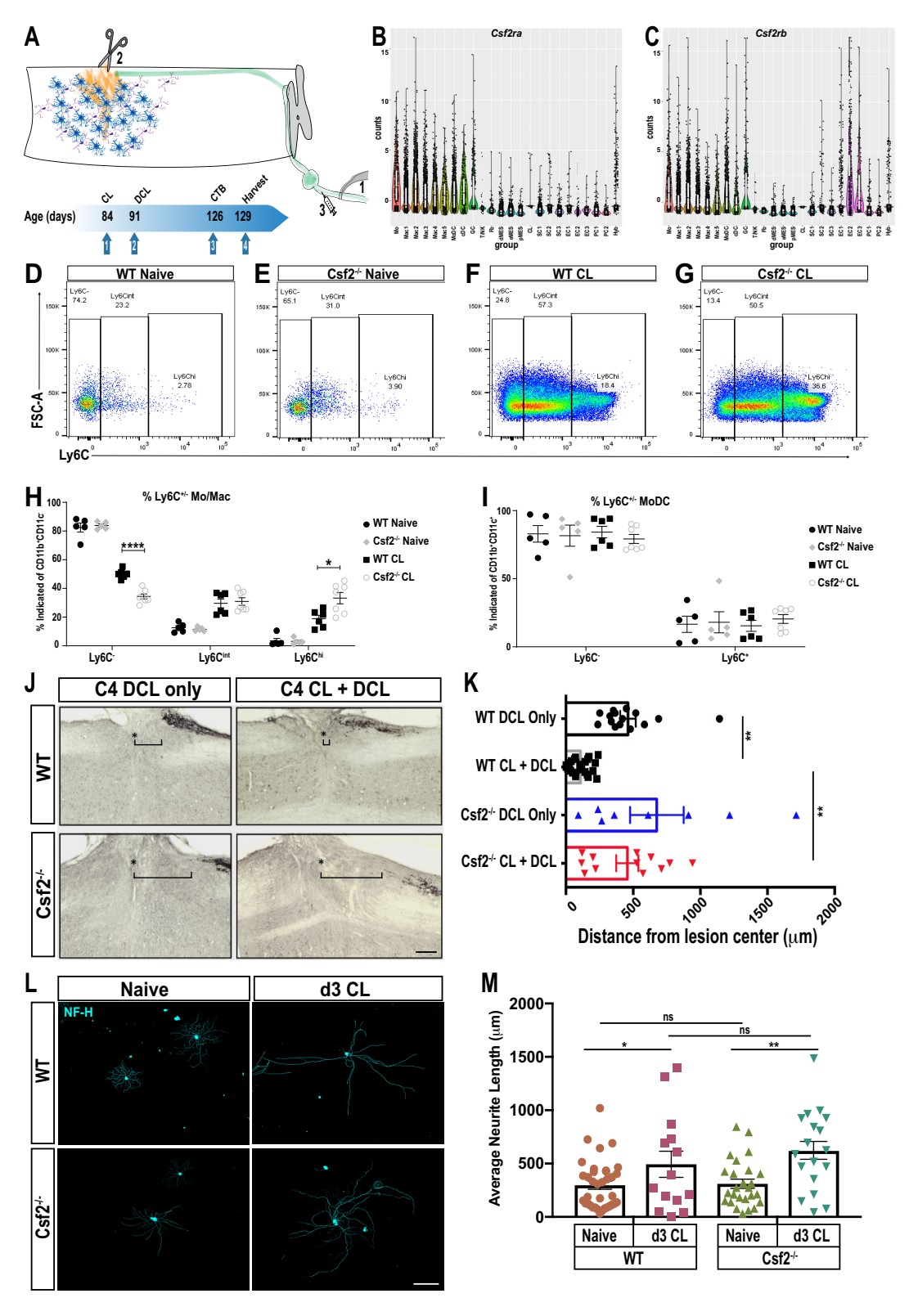

**Figure 8.** GM-CSF is required for conditioning lesion induced dorsal column axon regeneration. (A) Schematic showing conditioning lesion to the sciatic nerve (1) followed by dorsal column lesion (2) and tracer injection in the nerve (3). Experimental time line of conditioning lesion (CL), dorsal column lesion (DCL), cholera-toxin B (CTB) injection, and time of tissue harvest are shown. (B, C) Violin plots of *Csf2ra* and *Csf2rb* expression in the d3 post-SNC sciatic nerve, as assessed by whole nerve tissue scRNAseq analysis. (D–G) Flow cytometry dot plots of WT and *Csf2⁻/⁻* nerves from naive mice

*Figure 8 continued on next page*

*Figure 8 continued*

and 3d following conditioning lesion (CL) to the sciatic nerve. Ly6C surface staining was used to assess maturation of the Mo/Mac population. Ly6C$^{hi}$ (immature), Ly6C$^{int}$, and Ly6C$^-$ (mature) cells are shown. (H) Quantification of percentage of Mo/Mac (CD45$^+$ CD11b$^+$ CD11c$^-$ Ly6G$^-$) that are Ly6C$^-$, Ly6C$^{int}$ and Ly6C$^{hi}$ in WT and *Csf2$^{-/-}$* mice without (naive) and with CL. (I) Quantification of surface Ly6C on MoDC (CD45$^+$ CD11b$^+$ CD11c$^+$) in WT and *Csf2$^{-/-}$* mice without (naïve) and with CL. Unpaired t-test with correction for multiple comparisons using Holm-Sidak method, *p<0.05; ****p<0.0001. (J) Sagittal sections through cervical spinal cords of wild-type (WT) and *Csf2$^{-/-}$* mice, 5 weeks following bilateral DCL at cervical level 4 (C4). The spinal cord lesion site is labeled with a star (*), rostral is to the left and caudal is to the right. To enhance dorsal column axon regeneration, a CL to the sciatic nerve was performed 7 days prior to DCL (CL + DCL). Dorsal column axons were visualized by CTB injection in the sciatic nerve. The brackets indicated the distance between the lesion center and the rostral tip of CTB labeled axons. (K) Quantification of axon regeneration. The distance between CTB labeled axon tips and the center of the spinal lesion was measured; 0 µm marks the injury site, the gap between the lesion center and traced axons (=retraction) is shown for WT and *Csf2$^{-/-}$* without CL. For each genotype and experimental condition n ≥ 8 biological replicates. One-way ANOVA with Tukey posthoc correction. **p<0.01. Scale bar, 200 µm. (L) Representative images primary DRG neurons isolated from WT and *Csf2$^{-/-}$* mice, with and without a d3 CL. Cultures were stained with of anti-neurofilament H (NF-H) (M) Quantification of neurite length. Neuromath was used to quantify neurite length, neurites less than 30 µm in length were excluded from the analysis. n ≥ 114 neurons, n = 2 biological replicates. Two-tailed Student's t-Test with Tukey posthoc correction was used. *p<0.05; **p<0.01. Scale bar, 500 µm.

The online version of this article includes the following figure supplement(s) for figure 8:

**Figure supplement 1.** Locus and PCR genotyping of *Csf2$^{-/-}$* mice.

mice that received a conditioning lesion, traced axons grew close to the spinal cord injury site (*Figure 8J and K*). In parallel processed *Csf2$^{-/-}$* mice, without conditioning lesion, there is a 720 ± 120 µm gap between the lesion center, and the most proximal CTB-labeled axons (*Figure 8J and K*). However, in *Csf2$^{-/-}$* mice subjected to a conditioning lesion, dorsal column axon regeneration is not significantly enhanced (*Figure 8J and K*). This shows that *Csf2* is important for conditioning-lesion-induced central axon regeneration.

GM-CSF has pleiotropic functions and its receptors are found on hematopoietic cells, glial cells, and subsets of neurons (*Donatien et al., 2018*; *Franzen et al., 2004*). SNC leads to upregulation of GM-CSF in the nerve (*Mirski et al., 2003*) and acute administration of GM-CSF following SNC leads to a transient increase in PNS axon regeneration (*Bombeiro et al., 2018*). To assess whether loss of *Csf2* attenuates neurite outgrowth in vitro, we cultured DRG neurons from adult WT and *Csf2$^{-/-}$* mice. After 20 hr, many neurons with axons were identified in both WT and *Csf2$^{-/-}$* cultures (*Figure 8L*). Quantification of axon growth did not identify *Csf2* dependent differences in total axon length or the longest axon (*Figure 8M*). A second cohort of WT and *Csf2$^{-/-}$* mice was subjected to a conditioning lesion 3d prior to harvesting of axotomized DRGs. In both WT and *Csf2$^{-/-}$* cultures, neurite outgrowth is significantly increased when compared to DRGs prepared from naive mice (*Figure 8M*). Collectively, this shows that reduced axon regeneration in the dorsal columns of *Csf2$^{-/-}$* mice is not due to loss of conditioning-lesion-induced activation of neuron-intrinsic growth programs and indicates that *Csf2* promotes regeneration through cell non-autonomous, extrinsic mechanisms.

## Discussion

We show that compression injury to the sciatic nerve triggers massive infiltration of blood-borne immune cells into the nerve. Granulocytes enter first, closely followed by Ly6C$^{hi}$ monocytes. After a short pro-inflammatory phase, the immune milieu rapidly transitions toward resolution and is dominated by Ly6C$^-$ Mac. Analysis of axotomized DRGs revealed upregulation of immune-associated gene co-expression networks, however infiltration of blood-borne immune cells was very limited. DRG resident macrophages downregulate surface Ly6C upon nerve injury and undergo striking morphological changes. Single-cell RNA-seq identified 10 immune cell clusters in the injured nerve. Monocytes and their descendants, Mac1-Mac5 subpopulations and MoDC are abundantly present. The immune compartment includes a population of blood-derived, proliferating myeloid cells (Mac5) with stem-like features. Mononuclear phagocytes in the injured nerve form a connected continuum of 8 cell clusters, including a subpopulation of Arg1$^+$ Mac localized to the nerve crush site. In contrast, F4/80$^+$ Mac are more evenly distributed in the nerve and associated with Wallerian degeneration. Apoptotic cell corpses rapidly accumulate in the injured nerve. Experiments with parabiotic mice show that Mo/Mac and MoDC contribute to nerve debridement by 'eating' apoptotic leukocytes. In *Csf2$^{-/-}$* mice, pro-inflammatory Ly6C$^{hi}$ Mo/Mac are elevated in the injured nerve, while the

number of anti-inflammatory Ly6C⁻ cells is reduced. This exacerbation of inflammation correlates with loss of conditioning-lesion-induced central axon regeneration. Collectively, a comparative analysis of the immune response to PNS injury reveals striking differences in the inflammatory landscape between the nerve injury site, the degenerating nerve stump, and axotomized DRGs. Efferocytosis of apoptotic leukocytes is identified as a key mechanism of nerve debridement and inflammation resolution. Perturbed resolution of nerve inflammation, as observed in $Csf2^{-/-}$ mice, blocks conditioning-lesion-induced central axon regeneration.

## Evidence for specific immune compartments within the injured nerve

Traumatic PNS injury causes necrosis of SC, MES, and vasculature-associated cells at the nerve injury site. Disruption of the vasculature leads to endoneurial bleeding and tissue hypoxia. Necrosis is a violent form of cell death that disrupts the plasma membrane and leads to the release of intracellular damage-associated molecular patterns (DAMPs) into the extracellular milieu. Release of intracellular content, in any tissue, causes a strong pro-inflammatory response (*Frank and Vince, 2019*; *Vannella and Wynn, 2017*). Distal to the nerve crush site, transected nerve fibers undergo Wallerian degeneration and release DAMPs as they disintegrate. However, in the distal nerve the abundance and composition of DAMPs, such as the absence of double-stranded DNA and nuclear proteins, is very different from the nerve crush site (*Bortolotti et al., 2018*). Thus, depending on where Mo enter the injured nerve, they may encounter very different microenvironments and adapt site-specific phenotypes (*Canè et al., 2019*). The strong accumulation of *Arg1-YFP*⁺ cells at the nerve injury site, but not along degenerating fibers, supports the idea that Mo/Mac adapt microenvironment-specific phenotypes. Studies with chimeric mice show that hematogenous leukocytes first accumulate at the injury site and later along severed fibers that undergo Wallerian degeneration. The density of blood-derived leukocytes is highest at the injury site and correlates with the extent of tissue damage. We speculate that F4/80⁺ Mac associated with Wallerian degeneration function in phagocytosis of myelin debris and degenerated axons, whereas *Arg1*⁺ Mac near the injury site primarily function in removal of apoptotic cell corpses. In support of this idea, *Arg1*⁺ Mac, highly enriched in cluster Mac4, express the highest levels of engulfment receptors and gene products important for reverse cholesterol transport, a strong indicator for ongoing efferocytosis (*Yvan-Charvet et al., 2010*).

## Efferocytosis of apoptotic leukocytes in the injured sciatic nerve

Studies with chimeric mice show that upon sciatic nerve injury, Mo/Mac, and to a lesser extent MoDC, participate in nerve debridement by eating dying leukocytes. Bridging molecules that facilitate recognition of AC are abundantly expressed by immune and non-immune cells in the injured nerve. Compared to Mac from naïve PNS tissue, cells in subcluster Mac4 of the injured nerve show highly elevated expression of engulfment receptors. Some engulfment receptors, including *Lrp1*, *Axl*, and the scavenger receptor class B member 2 (*Scarb2*), are expressed by MES and repair SC, suggesting that immune and non-immune cells participate in nerve debridement, possibly including efferocytosis. Whether the large and diverse array of engulfment receptors expressed in the injured nerve reflects eating of specific debris, AC corpses, or a high degree of functional redundancy is unknown and requires further investigation. The most likely prey eaten by Mo/Mac and MoDC are dying neutrophils. Neutrophils are very abundant at early post-injury time points, have a short life span, and spontaneously die by apoptosis (*Greenlee-Wacker, 2016*; *Lindborg et al., 2017*). In non-neural tissues, efferocytosis of neutrophils triggers anti-inflammatory responses in Mo, Mac, and DC, a prerequisite for inflammation resolution (*Greenlee-Wacker, 2016*). Thus, efferocytosis is not simply a mechanism for garbage removal, but also a key driver to reprogram professional phagocytes from a pro-inflammatory to an anti-inflammatory state (*Boada-Romero et al., 2020*; *Eming et al., 2017*; *Ortega-Gómez et al., 2013*). In a similar vein, efferocytosis in the injured sciatic nerve may drive inflammation resolution and wound healing. In humans, dysregulation of efferocytosis can cause chronic inflammatory and autoimmune diseases, including asthma, systemic lupus erythematous, and atherosclerosis (*Kawano and Nagata, 2018*). Additional studies are needed to determine whether defective efferocytosis and impaired inflammation resolution in the PNS contribute to excessive tissue damage and neuropathic pain.

## The immune compartment of the 3-day injured sciatic nerve exhibits an immunosuppressive character

Rapid removal of AC corpses protects from secondary necrosis and is closely associated with the induction of immunological self-tolerance. Commensurate with this, the low presence of lymphocytes and Natural killer cells in the nerve indicates that the microenvironment is immunologically 'cold' and dominated by immunosuppressive mechanisms. We propose that efferocytosis in the injured nerve is key to switch from a pro-inflammatory environment to resolution and restoration of tissue integrity (*Kourtzelis et al., 2020*; *Ortega-Gómez et al., 2013*). At 3d post-SNC, expression of the pro-inflammatory cytokines and chemokines (*Ifng, Il1a, Il1b, Tnf*) is very low. Most myeloid cells express high levels of anti-inflammatory *Cd52*, a glycoprotein that binds to HMGB1 to suppress T cell function (*Bandala-Sanchez et al., 2018*; *Rashidi et al., 2018*). *Trem2*+*Arg1*+ cells are strongly enriched in cluster Macs4 and show gene signatures suggestive of myeloid-suppressive cells (*Katzenelenbogen et al., 2020*; *Yurdagul et al., 2020*). Further evidence for an immunosuppressive environment is the strong expression of *Pirb* by myeloid cells, a type one membrane protein with four cytoplasmic immunoreceptor tyrosine-based inhibitory motifs (ITIMs) that inhibit immune cell activation (*van der Touw et al., 2017*). Myeloid inhibitory C-type lectin-like receptor (*Clec12a), Lair1* (leukocyte-associated Ig-like receptor-1), *Fcgr2b* (low affinity immunoglobulin gamma Fc region receptor IIb), and the CD300 family receptors *Cd300a* and *Cd300lf*, all of which contain ITIMs (*Rozenberg et al., 2018*), are strongly expressed, and thus, may reduce nerve inflammation. TGFβ is expressed by efferocytotic Mac in the lung (*Yoon et al., 2015*). In the injured sciatic nerve, *Tgfb1* is expressed by myeloid cells and is important for axon regeneration (*Clements et al., 2017*; *Kourtzelis et al., 2020*). Cells in clusters Mac1, Mac2, and Mac3 express high levels of *Rbpj*, a TR that restrains ITAM (immunoreceptor tyrosine-based activation motif) signaling and promotes a, resolving Mac phenotype (*Foldi et al., 2016*). Mac4 cells express the transmembrane glycoprotein NMB (*Gpnmb*), a negative regulator of inflammation that has protective effects following tissue injury (*Zhou et al., 2017*). Of interest, in the 3d injured nerve, Mac1, Mac3, and Mac4 strongly express the TRs *Maf*/c-Maf and *Mafb/*MafB. MafB promotes reprogramming of macrophages into an M2-like, resolving phenotype (*Kim, 2017*) and c-Maf is a checkpoint that programs Mac and is critical for the acquisition of an immunosuppressive phenotype (*Liu et al., 2020*).

## *Csf2* deficiency alters nerve inflammation and blocks conditioning-lesion-induced axon regeneration

Parabiosis revealed massive infiltration of blood-borne immune cells into the injured nerve but not axotomized DRGs. This finding was independently confirmed by flow cytometry, Western blotting, 3D reconstruction of Iba1+ cells, and RNA-seq of axotomized DRGs. The small increase in hematogenous leukocytes in axotomized DRGs was unexpected, since infiltration of Mo/Mac is thought to be a key driver of conditioning lesion induced axon regeneration (*Kwon et al., 2015*; *Richardson and Issa, 1984*; *Zigmond and Echevarria, 2019*). Consistent with previous reports, sciatic nerve injury causes a strong increase in Iba1 immunoreactivity in DRGs. We provide evidence that increased Iba1 immunoreactivity is, at least in part, a reflection of macrophage morphological changes triggered by nerve injury. Additional mechanisms may include local myeloid cell proliferation (*Yu et al., 2020*) and infiltration of a small number of blood-borne myeloid cells.

SNC triggers an inflammatory response in the nerve and in axotomized DRGs, although quantitatively and qualitatively very different, it remains unclear which immune compartment is important for conditioning-lesion-elicited axon regeneration. To revisit this question, we took advantage of RNA-seq datasets generated from DRGs and nerves and searched for immune signaling pathways preferentially upregulated in the injured nerve, but not axotomized DRGs. Focusing on GM-CSF signaling, a cytokine that rapidly accumulates in the distal nerve stump (*Mirski et al., 2003*), we observed strong expression of both GM-CSF receptor subunits (*Csf2ra* and *Csf2rb*) in nerve macrophages but not axotomized DRGs. GM-CSF is known to promote Mo migration and Mac polarization (*Ijaz et al., 2016*; *Vogel et al., 2015*; *Wicks and Roberts, 2016*). Of interest, GM-CSF upregulates surface expression of galectin-3 on SC and Mac (*Saada et al., 1996*) and galectin-3 is thought to promote phagocytosis of myelin debris and participate in re-programming of Mac toward an anti-inflammatory phenotype (*Erriah et al., 2019*; *Rotshenker, 2009*). Recent evidence shows that galectin-3 promotes efferocytosis of neutrophils and promotes inflammation resolution (*Quenum Zangbede et al.,*

*2018*; *Wright et al., 2017*). Following SNC in *Csf2$^{-/-}$* mice, the ratio of Ly6C$^{hi}$ to Ly6C$^-$ Mo/Mac is significantly skewed toward the former. Functional studies with *Csf2$^{-/-}$* mice highlight a critical role for conditioning-lesion-induced regenerative growth of severed dorsal column axons. Neurite outgrowth studies with primary DRG neurons suggest that the regenerative failure in *Csf2$^{-/-}$* mice is not due to failed activation of DRG neuron intrinsic growth programs, but due to changes in extrinsic, environmental influences. Because *Csf2* receptor expression is very low in axotomized DRGs, this suggests that *Csf2*-dependent accumulation of Ly6C$^-$ Mac in the nerve is important for conditioning-lesion-induced axon regeneration. We speculate that *Csf2* functions non-cell autonomously in the injured nerve to generate an extracellular milieu capable to sustain neuron intrinsic growth programs activated by injury. In a similar vein, axotomy to corticospinal neurons is sufficient for the induction of neuron-intrinsic growth programs, but not maintenance. However, neuron-intrinsic growth programs in corticospinal neurons can be maintained by environmental cues released from stem cells grafted near the injury site (*Kumamaru et al., 2018*; *Poplawski et al., 2020*). While our studies demonstrate an important role for *Csf2* in conditioning-lesion-induced axon regeneration, we cannot rule out potential contributions by DRG macrophages. However, the small number of hematogenous macrophages detected in axotomized DRGs suggests that potential pro-regenerative immune mechanisms would need to be exerted by tissue resident macrophages. We acknowledge that axon regeneration was examined in *Csf2* global knock-out mice, and thus, it is possible that *Csf2* deficiency affects immune cells before they enter the injured sciatic nerve (*Hamilton, 2019*) or within the injured spinal cord (*Choi et al., 2014*; *Huang et al., 2009*).

Taken together, we provide a comparative analysis of SNC-induced inflammation in the nerve and axotomized DRGs and identify two very different immune compartments, the former primarily comprised of hematogenous leukocytes and latter of tissue resident endoneurial Mac. Mac subpopulations in the injured nerve are not uniformly distributed, indicating the existence of specific immune microenvironments. Efferocytosis of dying leukocytes is observed in the injured nerve, and thus, contributes to nerve debridement and inflammation resolution. If this process is curtailed, conditioning-lesion-induced regeneration of DRG neuron central axons is impaired.

# Materials and methods

## Key resources table

| Reagent type (species) or resource | Designation | Source or reference | Identifiers | Additional information |
|---|---|---|---|---|
| Antibody | Neurofilament heavy chain (chicken polyclonal) | Aves Lab | NFH | 1:750 |
| Antibody | Anti-chicken Cy3 (donkey polyclonal) | Jackson Immunoresearch | 703-165-155 | 1:200 |
| Antibody | Iba1 (rabbit polyclonal) | Wako Chemicals | 019–19741 | 1:500 |
| Antibody | F4/80 (Rat IgG2b monoclonal) | Thermo Fisher Scientific | ma1-91124 | 1:500 - 1:1000 |
| Antibody | CD68 (rabbit polyclonal) | Abcam | ab125212 | 1:500 |
| Antibody | GFAP (rabbit polyclonal) | DAKO | Z033429-2 | 1:500 |
| Antibody | SCG10 (rabbit polyclonal) | Novus Biologicals | NBP149461 | 1:500 - 1:1000 |
| Antibody | CTB (goat polyclonal) | List Biological Laboratories | 703 | 1:10,000 |
| Antibody | CD11b (rabbit monoclonal) | Abcam | ab133357 | 1:200-1:1000 |
| Antibody | ERK1/2 (rabbit polyclonal) | Cell Signaling | 9102 | 1:5000 |
| Antibody | Anti-rabbit HRP (donkey polyclonal) | EMD Millipore | AP182P | 1:2000-1:10000 |

*Continued on next page*

*Continued*

| Reagent type (species) or resource | Designation | Source or reference | Identifiers | Additional information |
|---|---|---|---|---|
| Antibody | CD16/32 (Rat-IgG2a monoclonal) | BD Pharmingen | 553141 | 1 μg / 1 million cells / 25 μL |
| Antibody | CD11b-PE-Cy7 (Rat-IgG2b monoclonal) | Thermo Fisher Scientific | 25-0112-82 | 1:200 |
| Antibody | Isotype Control-PE-Cy7 (Rat-IgG2b monoclonal) | Thermo Fisher Scientific | 25-4031-82 | 1:100 |
| Antibody | CD45-e450 (Rat-IgG2b monoclonal) | Thermo Fisher Scientific | 48-0451-82 | 1:100 |
| Antibody | Isotype Control-e450 (Rat-IgG2b monoclonal) | Thermo Fisher Scientific | 48-4031-82 | 1:100 |
| Antibody | CD45.1-e450 (Mouse-IgG2a monoclonal) | Biolegend | 110721 | 1:100 |
| Antibody | Isotype Control-e450 (Mouse-IgG2a monoclonal) | Biolegend | 400235 | 1:100 |
| Antibody | CD45.2-APC (Mouse-IgG2a monoclonal) | Biolegend | 109813 | 1:100 |
| Antibody | Isotype Control-APC (Mouse-IgG2a monoclonal) | Biolegend | 400221 | 1:100 |
| Antibody | Ly6G-APC-Cy7 (Rat-IgG2a monoclonal) | BD Biosciences | 560600 | 1:100 |
| Antibody | Isotype Control-APC-Cy7 (Rat-IgG2a monoclonal) | BD Biosciences | 552770 | 1:100 |
| Antibody | CD11c-PerCP-Cy5.5 (ArmHam-IgG monoclonal) | Thermo Fisher Scientific | 45-0114-82 | 1:100 |
| Antibody | Isotype Control-PerCP-Cy5.5 (ArmHam-IgG monoclonal) | Thermo Fisher Scientific | 45-4888-80 | 1:100 |
| Antibody | Ly6C-FITC (Rat-IgM monoclonal) | BD Biosciences | 561085 | 1:100 |
| Antibody | Isotype Control-FITC (Rat-IgM monoclonal) | BD Biosciences | 553942 | 1:100 |
| Antibody | Iba1 (goat polyclonal) | Novus Biologicals | NB100-1028 | 1:200 |
| Antibody | Anti-goat Alexa Fluor 488 (donkey polyclonal) | Jackson Immunoresearch | 705-545-147 | 1:200 |
| Chemical Compound | TOPRO pan-nuclear stain | Thermo Fisher Scientific | T3605 | |
| Chemical Compound | Fixable Viability Dye | Thermo Fisher Scientific | 65086614 | 1:500 |
| Chemical Compound | Proteinase K | New England Biolabs | P8107S | |
| Chemical Compound | 10 mM dNTP mix | Promega | C1141 | |
| Chemical Compound | 5X Green GoTaq Buffer | Promega | M791A | |
| Chemical Compound | GoTaq DNA polymerase | Promega | M3005 | |
| Chemical Compound | Buprenorphine | Par Pharmaceutical | NDC12496-0757-1 | |
| Chemical Compound | Ketamine | Par Pharmaceutical | NDC42023-115-10 | |
| Chemical Compound | Xylazine | Akorn | NDC59399-110-20 | |
| Chemical Compound | Isoflurane | McKesson Corporation | 667940172 | |
| Chemical Compound | Rhodamine-conjugated dextran MW 3,000 (Microruby) | Life Technologies | D-7162 | |
| Chemical Compound | Cholera toxin B (CTB) | Life Technologies | C34775 | |
| Chemical Compound | Puralube Eye ointment | Dechra | NDC-17033-211-38 | |
| Chemical Compound | N2 | Gibco | 17502048 | |
| Chemical Compound | N1 | Sigma | N6530 | |

*Continued on next page*

*Continued*

| Reagent type (species) or resource | Designation | Source or reference | Identifiers | Additional information |
|---|---|---|---|---|
| Chemical Compound | Leibovitz-15 (L-15) | Gibco | 21083–027 | |
| Chemical Compound | Penicillin/Streptomycin | Life Technologies | 15140–122 | |
| Chemical Compound | DMEM Ham's F-12 | Gibco | 10565–018 | |
| Chemical Compound | Fetal Bovine Serum | Atlanta Biologicals | S11550 | |
| Chemical Compound | Cytosine arabinoside | Sigma-Aldrich | C1768 | |
| Chemical Compound | Collagenase type 2 | Worthington Biochemical | LS004176 | |
| Chemical Compound | PBS without Calcium, Magnesium | Gibco | 10010023 | |
| Chemical Compound | Poly-L-lysine MW 70,000–150,000 | Sigma-Aldrich | P4707 | |
| Chemical Compound | laminin | Sigma-Aldrich | L2020 | |
| Chemical Compound | Paraformaldehyde | Sigma-Aldrich | 158127–500G | |
| Chemical Compound | Triton-X100 | Sigma-Aldrich | T8787 | |
| Chemical Compound | Bovine Serum Albumin (BSA) heat shock fraction V | Fisher Scientific | BP1600 | |
| Chemical Compound | Hoechst 33342 | Invitrogen | H3570 | |
| Chemical Compound | Tissue-Tek O.C.T. Compound | Electron Microscopy Sciences | 62550–01 | |
| Chemical Compound | β-Glycerophosphate | Sigma-Aldrich | G9422-100G | |
| Chemical Compound | Sodium Orthovanadate (Na3VO4) | Sigma-Aldrich | S6508-10G | |
| Chemical Compound | Protease inhibitor cocktail | Sigma-Aldrich | P8340-5ML | |
| Chemical Compound | DC Protein Assay Kit | Bio-Rad | 5000111 | |
| Chemical Compound | 2x Laemmli sample buffer | Bio-Rad | 1610737 | |
| Chemical Compound | β-Mercaptoethanol | EMD Millipore | 6010 | |
| Chemical Compound | Blotting-grade blocker | Bio-Rad | 1706404 | |
| Chemical Compound | SuperSignal West Pico PLUS Chemiluminescent Substrate | Thermo Fisher Scientific | 34580 | |
| Chemical Compound | SuperSignal West Femto Maximum Sensitivity Substrate | Thermo Fisher Scientific | 34095 | |
| Chemical Compound | WesternSure PREMIUM Chemi Substrate | Li-Cor Biosciences | 926–95000 | |
| Chemical Compound | Fixable Viability Dye eF506 | Thermo Fisher Scientific | 65-0866-14 | |
| Chemical Compound | TRIzol | Thermo Fisher Scientific | 15596026 | |
| Chemical Compound | Dispase | Sigma-Aldrich | D4693 | |
| Chemical Compound | Actinomycin D | Sigma-Aldrich | A1410 | |
| Chemical Compound | Percoll | Sigma-Aldrich | P4937 | |
| Chemical Compound | MACS buffer | Miltenyi | 130-091-376 | |
| Chemical Compound | Hanks balanced salt solution | Gibco | 14025092 | |
| Chemical Compound | Sucrose | Fisher Scientific | S5-500 | |
| Other | Pasteur pipette | VWR | 14673–010 | |
| Other | Flat Bottom Cell Culture Plates, 24-well | Corning | 3524 | |

*Continued on next page*

*Continued*

| Reagent type (species) or resource | Designation | Source or reference | Identifiers | Additional information |
|---|---|---|---|---|
| Other | Superfrost Plus Microscope Slides | FisherScientific | 12-550-15 | |
| Other | Zeiss Axio Observer Z1 | Zeiss | 491912-0049-000 | |
| Other | Zeiss Axiocam 503 mono camera | Zeiss | 426559-0000-000 | |
| Other | EC PlnN 10x objective | Zeiss | 420341-9911-000 | |
| Other | Motorized tissue homogenizer | RPI | 299200 | |
| Other | Fisher Scientific Sonic Dismembrator | Fisher Scientific | Model 500 | |
| Other | Refrigerated Centrifuge | Eppendorf | 5424R | |
| Other | Photospectrometer | Molecular Devices | SpectraMax M5e | |
| Other | PVDF membrane | EMD Millipore | IPVH00010 | |
| Other | LI-COR C-Digit | Li-Cor Biosciences | CDG-001313 | |
| Other | 70 µm cell strainer | Corning | 352350 | |
| Other | Ammonium-Chloride-Potassium (ACK) Lysing Buffer | Gibco | A1049201 | |
| Other | Clinical Centrifuge | Beckman Coulter | Allegra 6R | |
| Other | 40 µm filter | BD Falcon | 352340 | |
| Other | Myelin Removal Beads | Miltenyi | 130-096-731 | |
| Other | MidiMACS separator | Miltenyi | 130-042-302 | |
| Other | LS Columns | Miltenyi | 130-042-401 | |
| Other | Hemacytometer | Millipore Sigma | Z359629 | |
| Other | Chromium Next GEM Chip G | 10X Genomics, Inc | NC1000127 | |
| Other | 10X Genomic Chromium Controller | 10X Genomics, Inc | N/A | |
| Other | NovaSeq Illumina 6000 | Illumina | N/A | |
| Other | Cryostat | Leica Biosystems | CM3050S | |
| Other | 70 µm Cell Strainer | Corning | 352350 | |
| Other | Confocal Microscope | Nikon | C1 | |
| Other | Confocal Microscope | Leica Biosystems | SP8 | |
| Genetic Reagent (*Mus musculus* C57BL/6) | *Csf2-/-* | Jackson Laboratories | Stock No: 026812 | PMID:8171324 MGI: J:17978 |
| Genetic Reagent (*Mus musculus* C57BL/6) | *ROSA26-TdTom* | Jackson Laboratories | Stock No: 007576 | PMID:17868096 MGI: J:124702 |
| Genetic Reagent (*Mus musculus* C57BL/6) | CD45.1 | Jackson Laboratories | Stock No: 002014 | PMID:11698303 MGI: J:109863 PMID:11994430 MGI: J:109854 PMID:12004082 MGI: J:109853 |
| Genetic Reagent (*Mus musculus* C57BL/6) | Wildtype, WT | Taconics | B6NTac | |
| Genetic Reagent (*Mus musculus* C57BL/6) | *Arg1-eYFP* | Jackson Laboratories | Stock No: 015857 | PMID:17450126 MGI: J:122735 |
| Oligonucleotide | Csf2 Forward | Integrated DNA Technologies | N/A | 5'-GTGAAACACAAGTTACCACCTATG-3' |

*Continued on next page*

*Continued*

| Reagent type (species) or resource | Designation | Source or reference | Identifiers | Additional information |
|---|---|---|---|---|
| Oligonucleotide | Csf2 Reverse | Integrated DNA Technologies | N/A | 5'-TTTGTCTTCCGCTGTCCAA-3' |
| Oligonucleotide | Neomycin Forward | Integrated DNA Technologies | N/A | 5'-CTTGGGTGGAGAGGCTATTC-3' |
| Oligonucleotide | Neomycin Reverse | Integrated DNA Technologies | N/A | 5'-AGGTGAGATGACAGGAGATC-3' |
| Software | WIS-Neuromath | Weizmann Institute of Science | Version 3.4.8 | PMID:23055261 |
| Software | Image Studio Software | Li-Cor Biosciences | Version 5.2.5 | |
| Software | NovaSeq control software | Illumina | Version 1.6 | |
| Software | Real Time Analysis (RTA) software | Illumina | Version 3.4.4 | |
| Software | Cellranger | 10X Genomics, Inc | Version 3.1.0 | |
| Software | FACSDiva | BD Biosciences | Version 7 | |
| Software | FlowJo | FlowJo LLC | Version 10.6.2 | |
| Software | Seurat | Satija Lab - New York Genome Center | Version 3.1.2 | |
| Software | R | r-project.org | Version 3.6.2 | |
| Software | Slingshot | Bioconductor | Version 1.4.0 | |
| Software | Ranger | Comprehensive R Archive Network | Version 0.12.1 | |
| Software | Prism | GraphPad | Version 7 and 8 | |
| Software | Imaris | Bitplane | | |
| Software | Leica Application Suite (LAS X) | Leica | | |
| Software | Zen Application Software | Zeiss | Pro 3.8 | |
| Other | SomnoSuite | Kent Scientific | SS-01 | |
| Other | Povidone-Iodine Prep Pad | PDI Healthcare | B40600 | |
| Other | Alcohol Prep, Sterile, Md, 2 Ply | Covidien | 6818 | |
| Other | Fine Forceps Dumont #55 Dumoxel | Roboz Surgical Instrument | RS-5063 | |
| Other | 7 mm Reflex Wound Clips | Cell Point Scientific | 203–1000 | |
| Other | Micro Friedman Rongeur | Roboz Surgical Instrument | RS-8306 | |
| Other | McPherson-Vannas Micro Dissecting Spring scissors | Roboz Surgical Instrument | RS-5600 | |
| Other | COATED VICRYL (polyglactin 910) Suture | Ethicon | J463G | |
| Other | dumont #7 curved forceps | Fine Science Tools | 11271–30 | |
| Other | Miltex halsted mosquito forceps | Integra LifeSciences | 724 | |
| Other | Nanofil 10 µL syringe | World Precision Instruments | NANOFIL | |
| Other | 36 g beveled nanofil needle | World Precision Instruments | NF36BV-2 | |

*Continued on next page*

*Continued*

| Reagent type (species) or resource | Designation | Source or reference | Identifiers | Additional information |
|---|---|---|---|---|
| Other | Non-absorbable sutures | Ethicon | 640G | |
| Other | Absorbable sutures | Ethicon | J463G | |

## Animals

All procedures involving mice were approved by the Institutional Animal Care and Use Committee at the University of Michigan and Weill Cornell Medicine, and performed in accordance with guidelines developed by the National Institutes of Health. Adult (8–16 week-old) male and female mice on a C57BL/6 background were used throughout the study. Mice were housed under a 12 hr light/dark cycle with standard chow and water ad libitum. Mouse lines included, $Csf2^{-/-}$ (Jackson Laboratories, Stock No: 026812), *ROSA26-tdTom,* constitutively expressing membrane bound tdTomato in all cells (Jackson Laboratories, Stock No. 007576), CD45.1 (Jackson Laboratories, Stock No: 002014), and *Arg1-eYFP* reporter mice (Jackson Laboratories, Stock No: 015857).

## Genotyping of *Csf2* mice

Genomic (g) DNA was isolated from adult WT or $Csf2^{-/-}$ mice. Briefly, tissue samples were harvested and digested in lysis buffer (10 mM TrisHCl pH8, 25 mM EDTA, 0.1 M NaCl, 1% SDS) with Proteinase K overnight at 55°C. The following day, gDNA was extracted and resuspended in water. The following PCR primers were used: Csf2 forward 5'-GTGAAACACAAGTTACCACCTATG-3', Csf2 reverse 5'-TTTGTCTTCCGCTGTCCAA-3'; neomycin forward 5'-CTTGGGTGGAGAGGCTATTC-3', neomycin reverse 5'-AGGTGAGATGACAGGAGATC-3'. PCR parameters: 95°C for 2 min, (95°C for 1 min, 55°C for 30 s, 72°C for 20 s) repeated for 35 cycles, 72°C for 5 min.

## Surgical procedures

All surgeries were carried out under aseptic conditions. Mice were deeply anesthetized with a mixture of ketamine (100 mg/kg) and xylazine (10 mg/kg) or with isoflurane (5% induction, 2–3% maintenance, SomnoSuite Kent Scientific). Buprenorphine (0.1 mg/kg) was given pre-emptively and post-operatively.

### Sciatic nerve crush injury

For sciatic nerve surgery, thighs were shaved and disinfected with 70% ethanol (Covidien, 6818) and iodine (PDI Healthcare, B40600). A small incision, at mid-thigh, was made on the skin, underlying muscles separated, and the sciatic nerve exposed. For sham operated mice, the nerve was exposed but not touched. For SNC, the nerve was crushed for 15 s, using fine forceps (Dumont #55, Roboz Surgical Instruments, RS-5063). Skin was closed with 7 mm reflex wound clips (Cell Point Scientific, 203–1000).

### Doral column lesion

Spinal cord surgery was carried out as described previously (*Yoon et al., 2013*). Briefly, the C4 lamina was removed using micro-rongeurs (Roboz Surgical Instruments, RS-8306) under a stereomicroscope. The spinal column was exposed, and McPherson-Vannas Micro Dissecting Spring Scissors (Roboz Surgical Instruments, RS-5600) were inserted 1 mm deep. A hemisection of the dorsal spinal cord was carried out to transect all axons in the dorsal columns. The lesion was confirmed by probing with fine forceps. Next, dorsal muscle layers were closed using Perma-Hand Black sutures (5–0, Ethicon) and skin incisions were closed using coated Vicryl sutures (5–0, Ethicon, J463G).

### Axon tracing

For tracing of ascending sensory axons in the dorsal columns, tracer was injected into the sciatic nerve 5 weeks after SCI (*Yoon et al., 2013*). Briefly, the sciatic nerve was exposed at mid-thigh level and held in place using dumont #7 curved forceps (Fine Science Tools, 11271–30) and Miltex halsted mosquito forceps (Integra LifeSciences, 12460–174) to provide tension for the injection. Cholera

toxin B (CTB, List Biological Laboratories, #104, 1.5 µl of 1% solution in water) was injected into sciatic nerves using a Nanofil 10 µl syringe with a 36-gauge beveled needle (World Precision Instrument, NF36BV-2). The needle was removed ~30 s after injection to prevent backflow of fluid. Mice were sacrificed 3 days after tracer injection, spinal cords sectioned and stained as described (Yoon et al., 2013). Dorsal column lesion completeness was confirmed by absence of traced axons in transverse spinal cord sections rostral to the lesion. The distance between the lesion epicenter and the tip of traced axons was quantified by an investigator blinded with respect to mouse genotype and whether a conditioning lesion was applied or not.

## Parabiosis

Isochronic, same sex mice were housed in the same cage for at least 2 weeks prior to surgery. Mice were deeply anesthetized and their left or right sides shaved from just above the shoulder to below the knee. Eye ointment was applied to both mice to prevent drying. The skin was cleaned three times using ethanol and iodine pads before a unilateral skin-deep incision was made from the elbow to the knee on each animal. Skin fascia adjacent to the incision was peeled back using a pair of blunt forceps. Mice were joined at the knee and elbow joints using non-absorbable sutures by running the suture needle through the muscle just under each joint in both animals and completing the suture. Absorbable sutures were used to join the skin of each mouse around the shoulder and hindlimbs. 7 mm reflex wound clips were used to join the remainder of skin between the mice. Mice were allowed to recover for 3–4 weeks before further surgery.

## DRG cultures

Unilateral SNC was performed on adult mice 3 days prior to culture. The uninjured side was used as control. The dorsal spinal column from adult mice was exposed and the identity of lumbar DRGs established by counting vertebras from the hipbone (Sleigh et al., 2016). L3-L5 DRGs were dissected and harvested into L-15 with N2 (Gibco, 17502048) or N1 (Sigma-Aldrich, N6530) supplement on ice. DRGs were rinsed five times in L-15 with Penicillin/Streptomycin (Life Technologies, 15140–122) and minced in growth media (DMEM Ham's F-12, 10% FBS, 1X N2 or N1 supplement and 16 nM Cytosine arabinoside [Sigma-Aldrich, C1768]) with McPherson-Vannas Micro Dissecting Spring scissors. DRGs were digested in collagenase type 2 (10 mg/ml, Worthington Biochemical, LS004176) in $Ca^{2+}$, $Mg^{2+}$-free PBS (Gibco, 100010023) at 37°C for 20 min. Ganglia were dissociated by trituration using a fire polished Pasteur pipette, followed by centrifugation (5 min, 160 x g) and trituration in wash buffer (DMEM Ham's F-12, Gibco, 10565–018; 10% FBS, Atlanta Biologicals, S11550; 1% Penicillin/Streptomycin, Life Technologies, 15140–122) twice. Cells were plated in growth media at a density of 0.5 DRG per well in a 24-well plate (flat bottom plates, Corning, 3524) coated with poly-L-lysine 0.01% (MW 70,000–150,000) (Sigma-Aldrich, P4707) for 45 min at 37°C, followed by wash in $dH_2O$, dried and coated with 0.2 mg/mL laminin (Sigma-Aldrich, L2020). Cells were placed in a humidified incubator at 37°C, 5% $CO_2$ for 20 hr.

## Immunofluorescence staining

Primary DRG neuron cultures were fixed in 4% paraformaldehyde (PFA) in 1x PBS (Sigma-Aldrich, 158127) for 15 min at RT, followed by two brief rinses in PBS. Cells were permabilized in 0.3% Triton-X100 (Sigma, T8787) in PBS for 5 min at RT. Cells were incubated in blocking buffer, 2% FBS, 2% heat shock fraction V BSA (Fisher Scientific, BP1600), 0.3% Triton-x-100 in PBS for 1 hr. Cells were incubated with anti-Neurofilament heavy chain (NFH, 1:100; Aves Lab,) in blocking buffer overnight at 4°C and rinsed 3x in 0.3% triton-x-100 in PBS, 5 min each. Donkey anti-chicken Cy3 (1:200, Jackson Immunoresearch, 703-165-155) in blocking buffer was added for 45 min at room temperature. Cells were rinsed in PBS for 5 min. Hoechst 33342 (1:50,000 in PBS; Invitrogen, H3570) was added for 10 min at RT, followed by two washes in PBS. Cells were imaged on a Zeiss Axio Observer Z1 fitted with a Zeiss Axiocam 503 mono camera using the EC PlnN 10x objective. Single plane, tile scans were randomly acquired for each well. For immunofluorescence staining of neural tissues, mice were killed and perfused transcardially with ice-cold PBS for 2 min followed by ice-cold, freshly prepared 4% paraformaldehyde for 10 min. Spinal cord, sciatic nerves, and L4-L5 DRGs were collected and post-fixed in perfusion solution overnight. After that the solution was switched to 30% sucrose in PBS and tissues were kept at 4°C for at least 12 hr. Tissues were covered with tissue Tek (Electron

Microscopy Sciences, 62550–01) and stored at −80℃. Spinal cord sections and longitudinal sciatic nerve sections were cut at 12 µm and DRGs at 10 µm thickness using a cryostat (Leica Biosystems, CM3050S). Sciatic nerve and DRG sections were mounted on Superfrost⁺ microscope slides (Fisher Scientific, 12-550-15) and air dried for at least 12 hr. Spinal cord sections were stained in 24-well plates as free floating sections. The following primary antibodies were used, anti-Iba1 (1:500; WAKO, 019–19741), anti-F4/80 (1:500; Thermo Fisher Scientific, MA1-91124), anti-CD68 (1:500, Abcam, ab125212), anti-GFAP (1:500, DAKO, Z0334), anti-SCG10 (1: 2,000, Novus Biological, NBP1-49461), anti-CTB (1: 10,000, List Biological Laboratories, #703).

## Quantification of neurite outgrowth

Neurite lengths was quantified as described previously (*Robak et al., 2009*). Briefly, neurofilament-H stained cultures were used for neurite growth analyses. Only cells with neurites $\geq$ 30 µm were included in the analyses from randomly acquired tile scans using WIS-Neuromath (*Kalinski et al., 2019*).

## Whole mount DRG analysis

### Staining

Mice were subjected to unilateral SNC as described above. L4 DRGs from the uninjured (intact) and injured side were dissected and post-fixed in 4% PFA/PBS overnight at 4℃. For tissue clearing of DRGs, we used the iDISCO technique (*Bray et al., 2017*; *Renier et al., 2014*). Briefly, post-fixed samples were washed in 1x PBS and then dehydrated at room temperature with a series of 15 min washes with methanol in 0.05x PBS (20%, 40%, 60%, 80% and 100% vol/vol). Samples were bleached overnight with 5% $H_2O_2$ in 100% methanol at 4℃. The next day samples were rehydrated with a series of 15 min washes of methanol in 0.05x PBS + 0.2% Triton x-100 (80%, 40%, 20%, and 0% vol/vol). Samples were permeabilized in 1xPBS with 0.2% Triton X-100, 20% DMSO, and 0.3$M$ Glycine at 37℃ for 4 hr, followed by blocking with overnight incubation at 37℃ in 1xPBS with 0.2% Triton X-100, 10% DMSO, and 6% donkey serum. Samples were then washed twice for 1 hr in room temperature 1xPBS with 0.2% Tween 20 and 10 µg/ml heparin (PTwH). Then, samples were incubated with goat anti-Iba1 (1:200, Novus Biologicals, NB100-1028) in PTwH plus 5% DMSO and 3% donkey serum at 37℃ for 3 days. Samples were washed six times in PTwH: three washes for 15 min at room temperature, followed by two washes for 1 hr at 37℃ and last wash overnight at 37℃. Incubation with donkey anti-goat Alexa Fluor 488 (1:200, Jackson ImmunoResearch, 705-545-147) and the pan-nuclear stain TOPRO3 (Thermo Fisher Scientific, T3605) was performed in PTwH solution plus 3% donkey serum for 2 days at 37℃. Then, the six washes in PTwH were repeated as above, and the next day samples were processed for clearing. Samples were dehydrated in methanol/water series of 20%, 40%, 60%, and 80% vol/vol for one hour each at room temperature followed by two washes in 100% methanol for 30 min each. Samples were then incubated in 66% dichloromethane (DCM) and 33% methanol, followed by two incubations in 100% DCM for 30 min each. Finally, samples were cleared and stored in dibenzylether (DBE).

### Morphological analysis

For each cleared DRG, three different regions of interest were acquired on an inverted Nikon C1 confocal microscope at 60X using 0.25 µm z-steps. Image stacks were processed in ImageJ software for background subtraction (rolling ball radius of 10 pixels for Iba1 channel, and 20 for Topro3 signal), followed by mean filtering (1.5-pixel radius for Iba1 signal, and 2.0 for Topro3). Filtered images were then processed in Imaris software (Bitplane) to perform 3D surface rendering, and extraction of morphological characteristics (e.g. number of structures, cell, and processes volume). Iba1 immunoreactive cells were categorized based on morphological parameters: somal shape, branch number, and branch extension. Amoeboid cells were defined as having rounded somata of variable size with occasional short ramifications. Elongated cells exhibited an extended and regular rod shaped or arced somal morphology with only rare short branches. Stellate cells were clearly distinguished from the other cell types by having three or more elongated and curved branches.

## Density analysis

For estimation of total Iba1 density, whole cleared DRGs were imaged using 3D tile scanning at 20X on a Leica Sp8 confocal microscope. Alignment and stitching were performed with the Leica Application Suite X (LAS X). Images were pre-processed using LAS X Lightning detection package, and subsequently processed using Imaris software. To estimate the total density of Iba1 labeling within DRGs, 3D surface rendering of Iba1 was used, and the volume of reconstructions was normalized against the total volume of the corresponding whole DRG. Group size was based on previously published work (*Hollis et al., 2015*).

## Western blot analysis

Sciatic nerves and L3-L5 DRGs were dissected and lysed separately in radioimmunoprecipitation assay (RIPA) buffer (150 mM NaCl, 50 mM Tris, 1% NP-40, 3.5 mM sodium dodecyl sulfate, 12 mM sodium deoxycholate, pH 8.0) supplemented with 50 mM β-glycerophosphate (Sigma-Aldrich, G9422-100G), 1 mM $Na_3VO_4$ (Sigma-Aldrich, S6508-10G), and protease inhibitor cocktail (1:100, Sigma-Aldrich, P8340-5ML). Tissues were kept on ice, briefly homogenized with a motorized tissue homogenizer (RPI, 299200), and subjected to sonication (Fisher Scientific Sonic Dismembrator, Model 500) at 70% amplitude for 3 s. Tissue lysates were centrifuged at 15,000 rpm at 4°C for 10 min (Eppendorf, 5424R). The supernatant was transferred to a new tube and protein concentration was measured with a *DC* Protein Assay Kit (Bio-Rad, 5000111) using a photospectrometer at 750 nm (Molecular Devices, SpectraMax M5e). Samples were diluted with 2x Laemmli sample buffer (Bio-Rad, 1610737) containing 5% β-mercaptoethanol (EMD Millipore, 6010), boiled for 10 min at 100°C, and stored at −80°C for analysis. For SDS-PAGE, equal amounts of total protein (5–10 μg) were loaded per lane of a 15% gel. Separated proteins were transferred onto PVDF membrane (EMD Millipore, IPVH00010) for 2.5 hr at 200 mA in cold transfer buffer (25 mM TrisHCl, 192 mM Glycine, 10% Methanol). Membranes were blocked in 5% blotting-grade blocker (BioRad, 1706404) prepared in 1x TBS-T (TBS pH 7.4, containing 0.1% Tween- 20) for 1 hr at room temperature, and probed overnight at 4°C with the following primary antibodies diluted in 1x TBS-T with 3% BSA (Fisher Scientific, BP1600): α-CD11b (1:1000, Abcam, ab133357), α-ERK1/2 (1:5000, Cell Signaling Technologies, 9102). Horseradish peroxide (HRP)-conjugated α-rabbit secondary IgG (EMD Millipore, AP182P) were used. All HRP-conjugated secondary antibodies were diluted at half the dilution of the corresponding primary antibody in 3% BSA in 1x TBS-T, and the HRP signal was developed with various strengths of chemiluminescent substrates from Thermo Fisher Scientific (Pico Plus, 34580 or Femto, 34095) or from Li-COR Biosciences (926-95000). Protein band intensity was visualized and quantified in the linear range using LI-COR C-Digit (CDG-001313) and Image Studio Software (Version 5.2.5).

## Cell isolation for flow cytometry

Adult mice, naïve and at d1, d3, and d7 post-SNC were deeply anesthetized with a mixture of Xylazine and Ketamine and perfused transcardially with ice-cold phosphate-buffered saline (PBS) for 5 min. DRGs at lumbar levels L3-L5 were harvested and pooled in ice-cold PBS. Injured and uninjured sciatic nerves were dissected. From injured nerves, the proximal stump and the distal stump (including the injury site) were harvested and pooled separately. Similar sized segments from uninjured nerves were collected for comparison. In addition, spleen was harvested.

### Flow cytometry

To analyze immune cell profiles in dorsal root ganglia (DRG), sciatic nerves (SN), and spleen, mice were transcardially perfused for 5 min with ice-cold PBS to flush out all blood cells in circulation. The spleen was dissected, and splenocytes were passed through a 70-μm Falcon cell strainer (Corning, 352350). Red blood cells were lysed with Ammonium-Chloride-Potassium (ACK) lysing buffer. DRG and SN were harvested bilaterally. For analysis of DRGs (6 DRGs per mouse X 3–4 mice = 18–24 DRGs) and SN from 2 to 3 mice (2 SN per mouse x 2–3 mice = 4–6 SN) were pooled separately and used for one run. The collected nerve segments were cut into small pieces with microscissors and incubated in 1 ml collagenase (4 mg/ml Worthington Biochemical, LS004176) and dispase (2 mg/ml, Sigma-Aldrich, D4693) in PBS for 30–45 min at 37°C degrees in a 15 mL conical tube. Tissues were gently triturated with a P1000 pipette every 10 min. Next, samples were rinsed in DMEM with 10% FBS and spun down at 650 g for 5 min. This resulting pellet gently re-suspended in 1 mL of 27%

Percoll (Sigma Aldrich, P4937) in PBS. Then 3 ml of 27% Percoll were added to bring the final volume to 4 ml. Samples were spun at 900 g for 20 min in a clinical centrifuge (Beckman Coulter Allegra 6R). The top layers (with myelin and other debris) were carefully aspirated. The final 100 µl were resuspended in 1 ml of PBS with 2% FBS and filtered through a pre-washed 40 µm Falcon filter (Corning, 352340). Cells were pelleted at 650 g for 5 min at 4°C. Cells were labeled with fixable viability dye (Thermo Fisher Scientific, 65086614), blocked with αCD16/32 (BD Pharmingen, 553141), and stained with fluorescent antibodies and isotype controls. Immune cells (CD45$^+$) were further classified as myeloid (CD45$^+$CD11b$^+$), cDC (CD45$^+$CD11b$^-$CD11c$^+$Ly6G$^-$), MoDC (CD45$^+$CD11b$^+$-Ly6G$^-$CD11c$^+$), GC (CD45$^+$CD11b$^+$Ly6G$^+$CD11c$^-$), and Mo/Mac (CD45$^+$CD11b$^+$Ly6G$^-$CD11c$^-$). Data were acquired using a FACSCanto II (BD Biosciences) flow cytometer and analyzed with FlowJo software (Treestar) as described previously (*Baldwin et al., 2015*).

### Antibodies
CD11b-PE-Cy7 (Thermo Fisher Scientific, 25-0112-82), Rat IgGk Isotype Control-PE-Cy7 (Thermo Fisher Scientific, 25-4031-82) CD45-e450 (Thermo Fisher Scientific, 48-0451-82), Rat IgG2b Isotype Control-e450 (Thermo Fisher Scientific, 48-4031-82), CD45.1-e450 (Biolegend, 110721), Mouse IgG2ak Isotype Control-e450 (Biolegend, 400235), CD45.2-APC (Biolegend, 109813), Mouse IgG2ak Isotype Control-APC (Biolegend, 400221), Ly6G-APC-Cy7 (BD Biosciences, 560600), Rat IgG2a Isotype Control-APC-Cy7 (BD Biosciences, 552770), CD11c-PerCP-Cy5.5 (Thermo Fisher Scientific, 45-0114-82), Arm Ham IgG Isotype Control-PerCP-Cy5.5 (Thermo Fisher Scientific, 45-4888-80), Ly6C-FITC (BD Biosciences, 561085), Rat IgM Isotype Control-FITC (BD Biosciences, 553942). All antibodies were used at a working concentration of 1:100 except for CD11b (1:200).

### Statistics
Statistical analysis was performed in GraphPad Prism (v7) using paired or un-paired 2-tailed Student's t test, or 1-way or 2-way ANOVA with correction for multiple comparisons with Tukey's post-hoc test, as indicated in the figure legends. A p value < 0.05 (*) was considered significant. p<0.01 (**), p<0.001 (***), and p<0.0001 (****).

## Transcriptomics analysis, bulk RNA-seq of DRGs and scRNA-seq of sciatic nerves
For gene expression analysis of naive and axotomized DRGs, we carried out bulk RNA sequencing of L3-L5 ganglia harvested from naïve mice (n = 3), d1 (n = 3), d3 (n = 3), and d7 (n = 3) following bilateral SNC. For each data point, 18 ganglia were collected form three mice, pooled, flash frozen and lysed in Trizol solution for RNA extraction (*Chandran et al., 2016*). RNA-sequencing was carried out using TrueSeq RiboZero gold (stranded) kit (Illumina). Libraries were indexed and sequenced over two lanes using HiSeq4000 (Illumina) with 75 bp paired end reads. Quality control (QC) was performed on base qualities and nucleotide composition of sequences, to identify potential problems in library preparation or sequencing. Sequence quality for the dataset described here was sufficient that no reads were trimmed or filtered before input to the alignment stage. Reads were aligned to the latest Mouse mm10 reference genome (GRCm38.75) using the STAR spliced read aligner (version 2.4.0). Average input read counts were 58.8M per sample (range 53.4M to 66.2M) and average percentage of uniquely aligned reads was 86.3% (range 83.8% to 88.0%). Raw reads were filtered for low expressed genes and normalized by variance stabilization transformation method. Unwanted variation was removed by RUVSeq (1.20.0) with k = 1. Differentially expressed genes were identified using the bioconductor package limma (3.42.2) with FDR < 0.1 and the resulting gene lists were used as input for Ingenuity pathway analysis (Qiagen). Weighted gene co-expression network analysis was conducted using WGCNA R-package (ver 1.69). Soft thresholding power of 18 was used to calculate network adjacency. CutHeight of 0.3 was used to merge similar co-expression modules. Enrichment analysis for gene set was performed with GSEA (ver 2.2.2) using MsigDB (ver 7.0). Normalized enrichment score (NES) was used to assess enrichment of gene sets.

## Preparation of cells for scRNA-seq
Mice were transcardially perfused with ice-cold PBS for 5 min to flush out all blood cells in circulation. The sciatic nerve trunk was harvested and a segment that contains the injury site and the distal

nerve stump, up to the branch point of the tibial nerve, used for further processing. A minimum of three mice (six nerves) was used to obtain sufficient cells for analysis using the 10x Genomics platform. The collected nerve segments were cut into small pieces with microscissers and incubated in 1 ml PBS supplemented with collagenase (4 mg/ml Worthington Biochemical, LS004176), dispase (2 mg/ml, Sigma-Aldrich, D4693), and actinomycin D (45 µM, Sigma Aldrich, A1410) for 30–45 min at 37°C in a 15-mL conical tube. Tissues were gently triturated with a P1000 pipette every 10 min. Next, samples were rinsed in DMEM with 10% FBS and spun down at 650 g for 5 min before removing supernatant. The resulting pellet was gently re-suspended in 1 mL of 27% Percoll (Sigma Aldrich, P4937) in PBS. Then 3 ml of 27% Percoll were added to bring the final volume to 4 ml. Samples were spun at 900 g for 20 min with no brake in a clinical centrifuge (Beckman Coulter Allegra 6R). The top layers (with myelin and other debris) were carefully aspirated. The final 100 µl were resuspended in 1 ml of PBS with 2% FBS and filtered through a pre-washed 40 µm Falcon filter (Corning, 352340) with an additional 5 ml of PBS with 2% FBS. Cells were pelleted at 650 g for 5 min at 4°C. The supernatant was removed and the cell pellet resuspended in 180 µl of MACS buffer (Miltenyi, 130-091-376) diluted 1:20 in PBS (final bovine serum albumin [BSA] was 0.5%) and 10 µl of myelin removal beads were added (Miltenyi, 30-096-731). To remove all myelin debris, cells were incubated with myelin depletion beads for 15 min at 4°C with intermitted tapping. Cells were rinsed in 5 ml of MACS buffer, gently inverted several times and spun at 300 g for 10 min. Cells were separated from myelin beads using the MidiMACS separator (Miltenyi, 130-042-302) and LS columns (Miltenyi, 130-042-401). The flow through solution with the cells was centrifuged and the cells resuspended in 50 µl of Hanks balanced salt solution (Gibco, 14025092) supplemented with 0.04% BSA (Fisher Scientific, BP1600). The cell number and live/dead ratio was determined using propidium iodine labeling and a hemocytometer.

## 10x Genomics single-cell RNA-seq library preparation

For encapsulation of single cells with microbeads into nanodroplets, the Chromium Next GEM Single Cell 3' GEM Library and Gel Bead Kit v3.1 and Chromium Next GEM Chip G Single Cell Kit were used. Approximately 12,000 cells in a final volume of 43 µl were used for barcoding, using the 10X Genomics Chromium Controller. The library preparation of barcoded cDNAs was carried out in a bulk reaction, following instructions provided by the manufacturer. A small aliquot of the library was used for quality control with a bioanalyzer followed by library sequencing at the Advanced Genomics Core of the University of Michigan. The NovaSeq Illumina 6000 was used with an S4 flowcell, yielding 1.05 Billion reads (7–11% of the flowcell) (Individual samples ranging from 290 to 424 million reads). NovaSeq control software version 1.6 and Real Time Analysis (RTA) software version 3.4.4 were used to generate binary base call (BCL) formatted files.

## Data analysis

Raw scRNAseq data were processed using the 10x Genomics CellRanger softeware version 3.1.0. The CellRanger 'mkfastq' function was used for de-multiplexing and generating FASTQ files from raw BCLs. The CellRanger 'count' function, with default settings was used with the mm10 reference supplied by 10x Genomics, to align reads and generate single cell feature counts. Per sample, approximately 5800 cells with a median of 2507 genes per cell were obtained. CellRanger filtered cells and counts were used for downstream analysis in Seurat version 3.1.2 implemented in R version 3.6.2. Cells were excluded if they had fewer than 200 features, more than 7500, or the mitochondrial content was more than 25%. Reads from multiple samples were merged and normalized following a standard Seurat SCTransform integration pipeline (*Hafemeister and Satija, 2019*); mitochondrial mapping percentage was regressed out during the SCTransform normalization step. Principal component analysis was performed on the top 3000 variable genes and the top 30 principle components were used for downstream analysis. A K-nearest neighbor graph was produced using Euclidean distances. The Louvain algorithm was used with resolution set to. five to group cells together. Non-linear dimensional reduction was done using UMAP. The top 100 genes for each cluster, determined by Seurat's FindAllMarkers function and the Wilcoxon Rank Sum test, were submitted to version 11 of the string-db.org to determine functional enrichment; referred to as STRING analysis.

To model developmental trajectories of cells that comprise the mononuclear phagocyte system (MPS), the Bioconductor package, slingshot version 1.4.0 was used. The integrated Seurat object

was subset to include only MPS cells and slingshot was instructed to start from monocytes. The pseudo-time from the three slingshot constructed lineages were used in random regression forest to reveal the most influential genes, on pseudo-time. Random forests were implemented with the Ranger package of R from 1400 trees, 200 genes at each node, and the Gini index, 'impurity', measure for gene importance. The bulk RNA-seq and scRNA-seq data is available online in the Gene Expression Omnibus (GEO) database (GSE153762).

Cell identities, as defined above, were saved for the 3d injured nerve. Single-cell transcriptomes from YFP.pos and YFP.neg macrophage populations identified in naïve peripheral nerve tissue (*Wang et al., 2020*), were downloaded and given the label Mac_Naive. The log2 transformed raw counts of the 3d injured Mac1-5 and Mo as well as the Mac_Naive cells were subjected to batch correction using the ComBat function from the Bioconductor 'sva' package (*Leek et al., 2012*). Injured nerve Mo/Mac and naïve Mac made up the two batches and the following arguments were passed to ComBat: mod = NULL, par.prior = TRUE, mean.only = FALSE, prior.plots = FALSE. After batch correction each cell type and gene had a highly repeated minimum number near 0. To aid in plotting and determining 'percent expressed' this value was replaced with 0. The average expression for each gene and each cell type was calculated for the purpose of making dotplots. Any cell type with more than 85% zeros was not given a dot. The dots represent percent expressed by radius and average expression, scaled across cell type, by color.

## Acknowledgements

We thank Richard Zigmond and members of the Giger lab for critical reading of the manuscript. This work was supported by NIH T32 NS07222, Training in Clinical and Basic Neuroscience (AK), NIH T32 GM113900, Training Program in Translational Research (LH), the New York State Department of Health Spinal Cord Injury Program C33267GG (EH and RG), the Wings for Life Foundation (CY), the National Eye Institute (NEI), National Institutes of Health, R01EY029159 and R01EY028350 (BMS and RG), the Stanley D and Joan H Ross Chair in Neuromodulation fund (BMS), and the Dr Miriam and Sheldon G Adelson Medical Research Foundation (RK, DG, RG).

## Additional information

### Funding

| Funder | Grant reference number | Author |
|---|---|---|
| New York State Department of Health | C33267GG | Edmund Hollis<br>Roman J Giger |
| National Institutes of Health | T32-NS07222 | Ashley L Kalinski |
| National Institute of General Medical Sciences | T32-GM113900 | Lucas D Huffman |
| Wings for Life | fellowship | Choya Yoon |
| Dr. Miriam and Sheldon G. Adelson Medical Research Foundation | Program | Riki Kawaguchi<br>Daniel H Geschwind<br>Roman J Giger |
| Stanley D. and Joan H. Ross Chair in Neuromodulation fund | endowment | Benjamin M Segal |
| National Institutes of Health | R01EY029159 | Benjamin M Segal<br>Roman J Giger |
| National Institutes of Health | R01EY028350 | Benjamin M Segal<br>Roman J Giger |

The funders had no role in study design, data collection and interpretation, or the decision to submit the work for publication.

## Author contributions
Ashley L Kalinski, Data curation, Formal analysis, Funding acquisition, Investigation, Visualization, Writing - review and editing; Choya Yoon, Data curation, Formal analysis, Funding acquisition, Investigation, Methodology; Lucas D Huffman, Formal analysis, Funding acquisition, Investigation, Visualization, Methodology, Writing - review and editing; Patrick C Duncker, Formal analysis, Investigation, Visualization, Methodology; Rafi Kohen, Juan Sebastian Jara, Formal analysis, Investigation; Ryan Passino, Hannah Hafner, Investigation; Craig Johnson, Data curation, Software, Formal analysis, Visualization, Methodology; Riki Kawaguchi, Resources, Data curation, Software, Formal analysis; Kevin S Carbajal, Investigation, Methodology; Edmund Hollis, Benjamin M Segal, Resources, Formal analysis, Supervision, Funding acquisition, Methodology; Daniel H Geschwind, Resources, Data curation, Software, Formal analysis, Supervision, Funding acquisition; Roman J Giger, Conceptualization, Formal analysis, Supervision, Funding acquisition, Investigation, Writing - original draft, Writing - review and editing

## Author ORCIDs
Ashley L Kalinski (iD) https://orcid.org/0000-0001-7611-0810
Edmund Hollis (iD) https://orcid.org/0000-0002-4535-4668
Daniel H Geschwind (iD) https://orcid.org/0000-0003-2896-3450
Roman J Giger (iD) https://orcid.org/0000-0002-2926-3336

## Ethics
Animal experimentation: All animal research was approved by the University of Michigan School of Medicine and conducted under the IACUC approved protocol PRO00007948.

## Decision letter and Author response
Decision letter https://doi.org/10.7554/eLife.60223.sa1
Author response https://doi.org/10.7554/eLife.60223.sa2

# Additional files

## Supplementary files
• Transparent reporting form

## Data availability
The bulk RNA-seq and scRNA-seq data is available online in the Gene Expression Omnibus (GEO) database (GSE153762).

The following dataset was generated:

| Author(s) | Year | Dataset title | Dataset URL | Database and Identifier |
|---|---|---|---|---|
| Kalinski AL, Giger RJ | 2020 | axotomized DRGs and injured sciatic nerve | https://www.ncbi.nlm.nih.gov/geo/query/acc.cgi?acc=GSE153762 | NCBI Gene Expression Omnibus, GSE153762 |

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
