## [Decision Letter]

**Acceptance summary:**

The manuscript by Kalinski et al. uses a combination of approaches including flow cytometry, scRNA-sequencing, parabiosis and transgenic reporter mice to characterize the immune cells at the site of an injured nerve. The study provides insight into the preparatory contributions of inflammation towards the regeneration of injured nerve tissue. Different populations of immune cells are shown to differentially populate regions of the injured nerve supporting the idea that GM-CSF signaling in the injured peripheral nerve is necessary for axon regeneration in spinal column after a conditioning lesion.

**Decision letter after peer review:**

Thank you for submitting your article "Sciatic Nerve Injury Triggered Inflammation, Insights into Conditioning-Lesion Induced Axon Regeneration" for consideration by *eLife*. Your article has been reviewed by two peer reviewers, and the evaluation has been overseen by a Reviewing Editor and Satyajit Rath as the Senior Editor. The following individual involved in review of your submission has agreed to reveal their identity: Alyson Fournier (Reviewer #2).

The reviewers have discussed the reviews with one another and the Reviewing Editor has drafted this decision to help you prepare a revised submission.

Summary:

Kalinski et al. use a combination of approaches including flow cytometry, scRNA-sequencing, parabiosis and transgenic reporter mice to characterize immune cells in the injured sciatic nerve and DRG. The study examines mechanisms by which inflammation helps to condition tissue for regeneration. They identify different populations of immune cells which differentially populate regions of the injured nerve and provide support for the idea that GM-CSF signaling in the injured sciatic nerve is necessary for spinal column axon regeneration after a conditioning lesion. All of the reviewers noted the abundance of data collected and that the data was technically sound overall, however, several concerns were raised regarding the presentation and description of results.

Essential revisions:

1) The major issue with the manuscript is that the presentation and description of results is unwieldy and is relatively inaccessible to the reader as presented. The extensive molecular detail and the inclusion of discussion points that should be relegated to the Discussion section. This is best exemplified for the presentation of results in Figure 5 and Figure 6. The text should be less descriptive and more explanatory similar to how it is written in the last paragraph of the subsection “The cellular landscape of injured peripheral nerve tissue”. The title also suggests that the inflammation that is triggered by crush injury should give insights into conditioning-lesion induced axon regeneration. However, the manuscript seems to consist of two independent stories. The descriptive parts on: “the cellular landscape…”, “the immune compartment…” and “the cell type specific expression of engulfment…” are very elaborate. Because these parts are so detailed, it is not connected to the rest of the story anymore and it seems to be an independent “storyline”. It would likely create a better reading flow if the descriptive part on the single cell RNA sequencing data is reduced to a minimum.

2) The scRNAseq data provides transcriptomic information of the whole nerve on day 3 after crush injury. As mentioned in the subsection “The cellular landscape of injured peripheral nerve tissue”, injury induced expansion of the immune compartment peaks around d3 when analyzed by flow cytometry. However, to be able to properly identify the corresponding transcriptomic changes in the immune compartment it would be better to include a dataset of scRNAseq of naive uninjured peripheral nerves. This could be either own experimental data or published data from one of the three recent scRNA-seq studies of murine peripheral nerves.

3) Figure 2A – Iba1 and F4/80 expression is stated to be maximal at 3 days post-injury based on fluorescence microscopy. However, the background staining for both red and green channels appears to be higher for days 1-7 post-SNC when compared to sham. The staining appears more intense in days 1-7 post-crush relative to control, but perhaps a more robust quantification of the images with a correction for the differences in background staining between sections is needed. Moreover, an issue of tissue integrity or folding in the section used for day 3 post-crush may be creating an artifact in top left of image due irregularities in the tissue. The intense overlap in the signal for all three channels supports this concern. If possible, a different section from this group that does not possess this issue should be examined. Alternatively, higher resolution images may be used to verify that distinct cellular structures are stained.

4) In Figure 9F the image magnifications appear to differ. Were the representative images of WT taken at the same magnification as the CSF2^-/-^ mice since the apparent somatic size of WT is smaller than that of CSF2^-/-^ based on their NF-H fluorescence staining? This would then require correction of the subsequent neurite length quantification on panel G. If the images were indeed taken at the same magnification, do the size of the somas significantly differ between two conditions.

5) Although the finding that GM-CSF is required for conditioning lesion induced axonal regeneration is exciting, the data acquired with the *Csf2* KO mice is very minimal compared to the detailed analysis of the injured sciatic nerve. The mouse line induces a global knockout of GM-CSF, which provides a limitation to the interpretation of the data. This limitation should be emphasized further.

6) In the scRNA-seq dataset several mesenchymal clusters are identified. How was endoneurial vs. perineurial MES distinguished as claimed in the text. The location of these sub-clusters should be underlined by supporting evidence from literature references or by in situ visualization. Otherwise the statement should be adjusted.

7) Why was Erk1/2 used for normalization when analyzing the CD11b signal by W. Blot to corroborate the FACS data since the expression of ERK is affected by sciatic nerve injury?

8) Statistical: For the quantitative analysis of immune profiles in the deafferent DRG, the n numbers are not consistent. It is not possible to do a solid statistical test on n=3 vs. n=12 or n=14 so please increase the n for d1 and d3. For an unpaired t-test, the groups should have equal variance. When comparing more than 2 groups a one way ANOVA should be used. Please check your statistical testing for figures that include bar graphs with more than 2 groups (so naive-d1-d3-d7). When mentioning average or median values in the text, please add SD or SEM (in the subsection “The cellular landscape of injured peripheral nerve tissue”, for example).

---

## [Author Response]

Essential revisions:1) The major issue with the manuscript is that the presentation and description of results is unwieldy and is relatively inaccessible to the reader as presented. The extensive molecular detail and the inclusion of discussion points that should be relegated to the Discussion section. This is best exemplified for the presentation of results in Figure 5 and Figure 6. The text should be less descriptive and more explanatory similar to how it is written in the last paragraph of the subsection “The cellular landscape of injured peripheral nerve tissue”. The title also suggests that the inflammation that is triggered by crush injury should give insights into conditioning-lesion induced axon regeneration. However, the manuscript seems to consist of two independent stories. The descriptive parts on: “the cellular landscape…”, “the immune compartment…” and “the cell type specific expression of engulfment…” are very elaborate. Because these parts are so detailed, it is not connected to the rest of the story anymore and it seems to be an independent “storyline”. It would likely create a better reading flow if the descriptive part on the single cell RNA sequencing data is reduced to a minimum.

We have made substantial changes throughout the manuscript to shorten descriptive sections for the scRNA-seq analysis presented in Figures 5, 6, and 8. The description of non-immune genes (e.g. vasculature and fibroblast-like/MES cells) in Figure 5 has been shortened (200 words less than in the original version) and the presentation of myeloid cell gene expression (Figure 6) has been trimmed down. We removed Figure 8 entirely from the Results section (667 words) and moved a small part to the Discussion. In addition, we moved several descriptive sections. However, since the identification of engulfment receptors and bridging molecules expressed by specific cell types in the injured nerve is novel and forms the rationale for why we looked at efferocytosis (eating of dead cells) of blood-derived leukocytes in the injured nerve, we kept this section of the manuscript similar to the original version (Figure 7). Our work is the first documentation of ongoing efferocytosis in the injured sciatic nerve. Since efferocytosis is a key mechanism (in non-neural tissues) for inflammation resolution, we think that this is an important finding that needs to be clearly documented. To better reflect the work presented in the current study, we changed the title to “Efferocytosis of dying leukocytes in the injured sciatic nerve “. We improved the transition from nerve inflammation and efferocytosis to inflammation resolution and axon regeneration. We show that *Csf2* (GM-CSF) deficiency results in impaired inflammation resolution in the injured nerve and this is associated with defects in conditioning-lesion induced central axon regeneration.

2) The scRNAseq data provides transcriptomic information of the whole nerve on day 3 after crush injury. As mentioned in the subsection “The cellular landscape of injured peripheral nerve tissue”, injury induced expansion of the immune compartment peaks around d3 when analyzed by flow cytometry. However, to be able to properly identify the corresponding transcriptomic changes in the immune compartment it would be better to include a dataset of scRNAseq of naive uninjured peripheral nerves. This could be either own experimental data or published data from one of the three recent scRNA-seq studies of murine peripheral nerves.

Agreed, in the revised manuscript we now included a comparison to immune cell single-cell transcriptomes from naïve peripheral nervous system tissue. Specifically, we compare our 3-day injury scRNA-seq data set to naïve nerve macrophages data sets recently published by the Randolph lab. (Wang et al., 2020). The comparisons to naïve nerve macrophages are shown in Figure 6—figure supplement 6, Figure 7—figure supplement 1, and Figure 7—figure supplement 2.

3) Figure 2A – Iba1 and F4/80 expression is stated to be maximal at 3 days post-injury based on fluorescence microscopy. However, the background staining for both red and green channels appears to be higher for days 1-7 post-SNC when compared to sham. The staining appears more intense in days 1-7 post-crush relative to control, but perhaps a more robust quantification of the images with a correction for the differences in background staining between sections is needed. Moreover, an issue of tissue integrity or folding in the section used for day 3 post-crush may be creating an artifact in top left of image due irregularities in the tissue. The intense overlap in the signal for all three channels supports this concern. If possible, a different section from this group that does not possess this issue should be examined. Alternatively, higher resolution images may be used to verify that distinct cellular structures are stained.

We agree with the reviewer and have replaced all panels in Figure 2A with images of tissue sections that are not damaged. The background in these images was corrected and the magnification increased. While there is extensive overlap between the anti-Iba1 and anti-F4/80 staining, the staining patterns are not identical. This can now be appreciated by the presence of “red only” and “green only” structures in merged images.

4) In Figure 9F the image magnifications appear to differ. Were the representative images of WT taken at the same magnification as the CSF2^-/-^ mice since the apparent somatic size of WT is smaller than that of CSF2^-/-^ based on their NF-H fluorescence staining? This would then require correction of the subsequent neurite length quantification on panel G. If the images were indeed taken at the same magnification, do the size of the somas significantly differ between two conditions.

In Figure 9F (now Figure 8L in the revised manuscript) we show primary DRG neurons prepared from WT and *Csf2* KO mice (with and without conditioning lesion); we double-checked the magnifications used when taking these images. They are all the same. The soma size of the DRG neurons does not change between WT and KO cultures, however the soma size is variable among different DRG neuronal populations.

5) Although the finding that GM-CSF is required for conditioning lesion induced axonal regeneration is exciting, the data acquired with the Csf2 KO mice is very minimal compared to the detailed analysis of the injured sciatic nerve. The mouse line induces a global knockout of GM-CSF, which provides a limitation to the interpretation of the data. This limitation should be emphasized further.

We agree with the reviewers that the use of a global *Csf2* global KO mouse does not allow us to pin-point where exactly GM-CSF (the protein encoded by *Csf2*) is required for conditioning-injury induced central axon regeneration. However, given the strong expression of the *Csf2* receptors (*Csf2ra* and *Csf2rb*) in the injured nerve, but not in axotomized DRGs, our work suggests that the nerve is the main site of action. In the revised Discussion, we further emphasized the limitations of using a global KO mouse for our studies. We discuss the possibility that *Csf2* deficiency may affect immune cells before they enter the injured sciatic nerve or exert effects in the central nervous system, which we have not examined in the present study. See the subsection “*Csf2* deficiency alters nerve inflammation and blocks conditioning lesion induced axon regeneration”

6) In the scRNA-seq dataset several mesenchymal clusters are identified. How was endoneurial vs. perineurial MES distinguished as claimed in the text. The location of these sub-clusters should be underlined by supporting evidence from literature references or by in situ visualization. Otherwise the statement should be adjusted.

Identification of different MES clusters was reported by Carr et al. 2019, after digit tip amputation in the mouse, combined with scRNA-seq. The original reference (Carr et al., 2019) describing the three MES clusters was cited in the original version of the manuscript, however for clarification, we now added the Carr et al., 2019 reference at the beginning of the description of the MES clusters.

7) Why was Erk1/2 used for normalization when analyzing the CD11b signal by W. Blot to corroborate the FACS data since the expression of ERK is affected by sciatic nerve injury?

Normalization of Western blots (WB) shown in Figure 2H: For normalization of samples we used total protein amount loaded per lane on the SDS-PAGE. While we agree, there are no perfect “loading controls”, total Erk1/2 is a commonly used loading control marker (e.g. PMID: 31068376) for WBs. Also, I’d like to point out that the observed differences in inflammation between the injured nerve and axotomized DRGs are based on multiple lines of evidence. In addition to WBs shown in Figure 2H, we show a detailed analysis by 1) Flow cytometry of nerve and DRGs, 2) RNA-sequencing, 3) immunofluorescence staining of DRG sections, and 4) 3D reconstruction of individual macrophages in axotomized DRGs. These different experimental approaches all point to the same conclusion: following sciatic nerve crush injury, inflammation in the nerve is orders of magnitude higher than in axotomized DRGs.

8) Statistical: For the quantitative analysis of immune profiles in the deafferent DRG, the n numbers are not consistent. It is not possible to do a solid statistical test on n=3 vs. n=12 or n=14 so please increase the n for d1 and d3. For an unpaired t-test, the groups should have equal variance. When comparing more than 2 groups a one way ANOVA should be used. Please check your statistical testing for figures that include bar graphs with more than 2 groups (so naive-d1-d3-d7). When mentioning average or median values in the text, please add SD or SEM (in the subsection “The cellular landscape of injured peripheral nerve tissue”, for example).

We appreciate the point raided by the reviewer regarding the statistical analysis of flow data of day 1 axotomized DRGs (Figure 2). While we agree with the reviewer that an n = 3 for the day 1 time-point is lower than for all the other time points shown, we did not do additional flow experiments at day 1 because there is no increase in DRG inflammation at day 1. We reached this conclusion based on multiple independent experiments, including bulk RNA-sequencing of naïve and axotomized DRGs, anti-CD11b WB analysis. We discussed this with the reviewing editor and it was concluded that additional flow experiments at day 1 are not necessary to conclude that inflammation in axotomized DRGs is not significantly elevated. Statistical methods for comparing more than 2 groups have been revised.